# The mouse multi-organ proteome from infancy to adulthood

Qingwen Wang[1,2], Xinwen Ding[1,2], Zhixiao Xu[1,2], Boqian Wang[1,2], Aiting Wang[1,2], Liping Wang[1], Yi Ding[1,2], Sunfengda Song[1,2], Youming Chen[1,2], Shuang Zhang[1,2], Lai Jiang[1] & Xianting Ding ®[1,2] ✉

The early-life organ development and maturation shape the fundamental blueprint for later-life phenotype. However, a multi-organ proteome atlas from infancy to adulthood is currently not available. Herein, we present a comprehensive proteomic analysis of ten mouse organs (brain, heart, lung, liver, kidney, spleen, stomach, intestine, muscle and skin) at three crucial developmental stages (1-, 4- and 8-weeks after birth) acquired using data-independent acquisition mass spectrometry. We detect and quantify 11,533 protein groups across the ten organs and obtain 115 age-related differentially expressed protein groups that are co-expressed in all organs from infancy to adulthood. We find that spliceosome proteins prevalently play crucial regulatory roles in the early-life development of multiple organs, and detect organ-specific expression patterns and sexual dimorphism. This multi-organ proteome atlas provides a fundamental resource for understanding the molecular mechanisms underlying early-life organ development and maturation.

Organ development and maturation are complicated processes involving dynamic alternation and precise orchestration of thousands of molecules, especially during adolescence which is often considered as the golden fitness time of life[1,2]. Understanding the developmental regulatory mechanisms in various organs during this period is thus especially important for decoding fundamental biological questions such as aging, health and diseases[3,4]. With the ultimate goal of defining patterns of organ development and maturation, numerous efforts have been made to comprehensively annotate the entire genomic atlases at the DNA, RNA, and protein levels[5–7]. To date, the molecular profiling of early-life multi-organ comparisons still remains scarce.

The development of an organism comprises numerous organs and tissues, and hence the interconnections between organs should be considered while investigating development[8,9]. Single-organ developmental studies have been reported[10–15] to understand individual organ functions. Yet, the vital inter-organ interactions, which encompass developmental synchronicity and heterogeneity, necessitate a comprehensive multi-organ atlas derived from individual projects. Such challenges could hardly be tackled by analyzing the commonalities or characteristics of multi-organ holistic developmental biology with integrated data from different studies because specific data acquisition methods and platforms used in different laboratories cannot guarantee to generate reproducible results[16,17]. The importance of parallel studies of multiple organs has been noted in research areas including development, evolution and aging, but has mainly focused on the RNA level[5,18]. Nevertheless, proteins participate more directly in a variety of biological activities and there is growing consensus that protein and RNA levels are commonly not conserved[19,20]. Mapping the multi-organ developmental proteome atlas can therefore detect the molecular regulators and effectors of physiological activities in developmental biology. However, our knowledge of the proteomes of multi-organ development and maturation remains inadequate.

Over the past decades, liquid chromatography coupled with tandem mass spectrometry (LC-MS) based proteomics methods have emerged into the mainstream for measuring the states of functional phenotypes[21,22]. Label-free quantification methods are commonly used due to experimental robustness and the ability to handle large groups

[1]Department of Anesthesiology and Surgical Intensive Care Unit, Xinhua Hospital, School of Medicine and School of Biomedical Engineering, Shanghai Jiao Tong University, Shanghai, China. [2]State Key Laboratory of Oncogenes and Related Genes, Institute for Personalized Medicine, Shanghai Jiao Tong University, Shanghai, China. ✉e-mail: dingxianting@sjtu.edu.cn

of samples[23]. Data-independent acquisition (DIA), one of the acquisition schemes of tandem mass spectrometry, offers high reproducibility, reduces random elements, and improves the accuracy of quantitative proteomics[22,24]. Instead of selecting specific precursors with higher intensities for fragmentation as in data-dependent acquisition (DDA), the mass spectrometer in DIA method is configured to cycle through a pre-defined set of precursor isolation windows, ensuring that all precursors within the desired mass range are consistently fragmented[21,25,26]. Recently, deep learning has been extensively used in DIA analysis as deep networks are able to exploit the intrinsic joint distribution of signals in the high-dimensional feature space[21,27]. The library-free DIA approach coupled with deep learning data analysis jointly benefits high-throughput applications and enhances the detection and quantification performance in traditional DIA proteomic applications[21,28].

In this study, we adopted the DIA method to examine differential protein expression in ten organs from both male and female mice at 1-, 4- and 8-weeks of age after birth, corresponding to the development from infancy to adulthood[29,30]. A ten-organ developmental proteomic atlas including 11,533 protein groups (PGs) was generated. We show that spliceosome proteins are not only extensively involved in the early-life development of multiple organs, but also have noteworthy modulatory functions for the organ-specific expression and sexual dimorphism during early-life liver development. This dataset serves as a valuable resource to facilitate further analysis of progressive organ development in mice from infancy to adulthood, as well as understanding the orchestration of organ development. The mass spectrometry proteomics data and searching output data are available via ProteomeXchange with the identifier PXD041400.

## Results

### A proteome atlas of ten mouse organs from infancy to adulthood

In order to investigate the dynamic protein variation in mice organs from infancy to adulthood, we conducted proteomic analysis on 10 organs (brain, heart, lung, liver, kidney, spleen, stomach, intestine, muscle and skin) isolated from C57BL/6JN male and female mice at postnatal weeks 1, 4 and 8 (5 mice for each sex at each age) (Supplementary Fig. 1a). These three time points, correspond to the development from infancy to adulthood, and represent critical stages in early-life multi-organ development[29,30]. Filter-Aided Sample Preparation (FASP)[31] workflow was applied to prepare a total of 300 organ samples (10 organs × 3 time points × 2 sex × 5 biological replicates) (Fig. 1a). All samples were measured on a quadrupole orbitrap mass spectrometer in 120 min DIA mode (Q Exactive HF-X, Methods). DIA-NN software (version 1.8)[21], which exploits deep neural networks, new quantification and signal correction strategies, was adopted to process our DIA raw data.

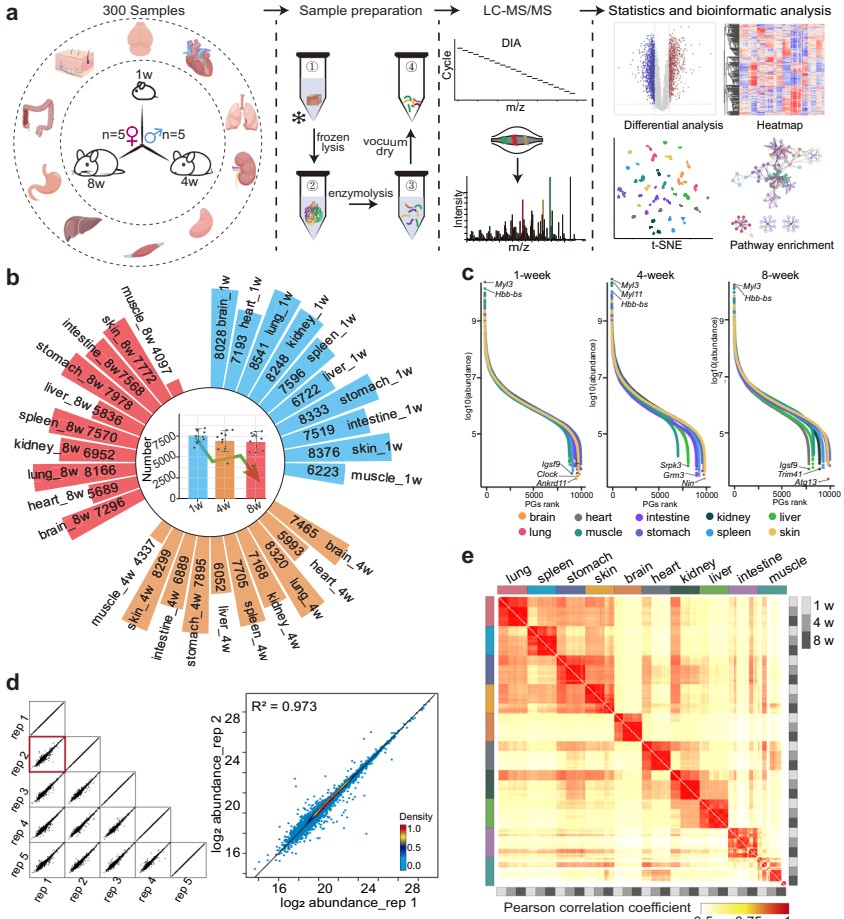

**Fig. 1 | Proteome atlas of ten mouse organs from infancy to adulthood. a** The workflow for DIA quantitative proteomics and bioinformatic analysis. Images of organs were generated by Figdraw, from https://www.figdraw.com. **b** The total number of protein groups (PGs) detected in each organ at each time point. The middle bar graph indicates the number of proteins detected in 10 organs at the same age. (Data are mean ± SD). **c** Dynamic ranges of the ten-organ proteomes.

**d** Left: multi-scatter plots comparing replicate samples. Right: scatter plots comparing two brain replicates, namely sample 1 and sample 2. **e** The Pearson correlation coefficient of all samples. Quality control analysis by Pearson correlation coefficient rank was applied to test the hypothesis that the protein correlation with disparate organs should be smaller than the correlation with the same organ at different time points. Source data are provided as a Source Data file.

A total of 11,556 protein groups (PGs) were detected with a false discovery rate (FDR) less than 1% at both peptide and protein levels. A final matrix consisting of 11,533 PGs was acquired for bioinformatic analysis after retaining PGs that were detected in at least four out of five biological replicates. (Supplementary Data 1). Figure 1b presents the number of PGs detected in ten organs at three different time points, with the range varying from 4097 (muscle_8-week) to 8541 (lung_1-week). The number of PGs detected in nine out of ten organs showed a decrease with development, with the exception of the intestine, where the number initially decreased and then increased from 1-week to 8-week (Supplementary Fig. 1b).

The logarithmic transformed quantitative data in this study span approximately ten orders of magnitude, which implies the highly dynamic nature of the proteome (Fig. 1c, Supplementary Data 2). The correlation between the biologically replicated samples of organs of the same sex averaged 0.974, which indicates the stability of the data (Fig. 1d Supplementary Data 3). To check for batch effects or run-to-run variations, we performed the following analyses. An initial PCA analysis was conducted on all samples without any sample or protein filtering. As expected, the samples exhibited clustering patterns based on organ (Supplementary Fig. 1c). According to previous studies on batch effects by Jelena et al.[32], we also examined the data after performing a log2 transformation and found no significant differences among the batches (Supplementary Fig. 1d). The number of proteins identified by QC was stable, showing that no effect along measurement time on instrument performance was observed (Supplementary Fig. 1e). Moreover, in order to investigate the potential impact of the sequencing order in mass spectrometry studies, a total of 40 samples from the brain, kidney, liver, and spleen were analyzed using random mass spectrometry. We found that regardless of the order or disorder of LC-MS measurement, the samples tightly clustered according to tissue types, indicating that in our study the detection order does not affect the mass spectrometry data or cause batch effects (Supplementary Fig. 1f, Supplementary Data 4). The coefficient of variation (CV) is used to assess the relative variability of the dataset, with higher CV values representing more dispersion[33]. In our study, protein quantification in most biological replicate samples showed a median coefficient of variance (CV) below 0.3 (Supplementary Fig. 2, Supplementary Data 5). Due to individual variation, the intestinal and muscle samples exhibited slightly higher coefficients of variation (CVs).

To further verify the accuracy of our proteomic data, we first verified randomly selected proteins from the Housekeeping gene pool[34] and confirmed that they were stably expressed across time and organs (Supplementary Fig. 1g). We also aimed to compare our findings with other existing studies on the proteomes of mouse single-organ[11–13]. Our data faithfully cover the majority of the PGs from studies of single organ proteomes (Supplementary Fig. 1h, Supplementary Data 1), with a total of 70% repetition in the lung study[13], 65% duplication in the liver study[11] and 55% duplication in the stomach study[12].

Inter-organ correlation studies are valuable for the investigation of systemic diseases. To detect the correlation between individual developmental age and organ, we performed Pearson correlation analysis on all samples (Fig. 1e, Supplementary Data 1). The brain demonstrates distinctiveness from other organs. Interestingly, the proteomic data from four organs, namely the lungs, spleen, stomach, and skin, demonstrate a higher degree of correlation, with an average Pearson correlation coefficient of 0.851. Inter-age studies found that with the exception of intestine, muscle and skin, 7 of the 10 organs showed higher Pearson correlations between 4-week and 8-week than the Pearson correlations between 1-week and 4-week or between 1-week and 8-week (Supplementary Fig. 3, Supplementary Data 3). For brain, the correlation between 1-week and 4-week, 1-week and 8-week, 4-week and 8-week were 0.76, 0.70 and 0.95, respectively. Contrary to other organs, skin at 1-week and 4-week has the highest correlation of 0.962, suggesting that the two states are closer to each other. This

finding is in line with the observations made by Dekoninck et al. in the murine tail and paw epidermis, which reported that the surface area of the tail experienced linear growth from postnatal day 1 to day 30, and then gradually reached a plateau[35]. Inter-age correlations are inconsistent across organs, indicating heterogeneity in the rate of development between organs.

In summary, our study provides a comprehensive data resource for understanding the molecular mechanisms of inter-organ development and maturation from infancy to adult. We next analyzed organ co-expressed PGs, multi-organ developmental synchronization, organ-unique PGs, age-unique PGs and sexual dimorphic PGs, respectively below.

## Age-related differential proteins across ten organs

To acquire age-related differential protein groups across different organs, we performed an ANOVA on the proteomic data of each organ separately by age (Supplementary Fig. 4, Supplementary Data 6). The largest number of differentially expressed protein groups (DEPs) (ANOVA adjust $p$-value < 0.05) were found in the lung as 5455, followed by skin, kidney, spleen and brain (Fig. 2a). KEGG enrichment analysis revealed that signaling pathways such as spliceosome, oxidative phosphorylation and insulin signaling were major enriched age-related DEPs in all ten organs (Fig. 2b, Supplementary Data 7). In addition, there are also age-related differential signaling pathways that belong exclusively to one or some organs. This observation characterizes the diversity and interplay of regulatory mechanisms implicated in the development of distinct organs of the organism.

To derive DEPs associated with whole organism development and to uncover their biological significance, the co-expressed PGs among 10 organs were further examined. Cross-referencing of DEPs from 10 organs detected 115 age-related developmental PGs present in all organs (Supplementary Data 8). The t-distributed stochastic neighbor embedding (t-SNE) clustering of the 115 DEPs showed that all the samples were correctly clustered by organ type and age stage (Fig. 2c). Heatmap of the 115 DEPs were differentially expressed in diverse organs (Fig. 2d). MFuzz analysis showed that 68 out of 115 DEPs were upregulated during 10 organs development, while 47 DEPs were downregulated during organ development (Fig. 2e). We analyzed the 115 DEPs as a whole because considering both up-regulated and down-regulated proteins together could potentially facilitate the identification of overarching regulatory patterns and interplay within biological processes. 96 out of 115 DEPs could form a protein inter-actions network (Fig. 2f). KEGG enrichment results shown these inter-organ age-related proteins extensively engaged in the metabolism of RNA. String analysis revealed that a remarkable 86 out of 96 proteins exhibited whole-body expression, implying that these 115 DEPs we propose have the potential to be applied to developmentally relevant differential proteins throughout the whole body from infancy to adulthood.

The enrichment analysis suggested that the 115 DEPs are mainly engaged in the metabolism of RNA and RNA splicing biological processing. The fact that only one KEGG pathway, the spliceosome, had an adjusted $p$-value of fewer than 0.05 highlights the important role of the spliceosome in organismal development and maturation (Fig. 2g). Spliceosome is mainly responsible for splice precursor messenger RNA (pre-mRNA), which is an extremely important step in the flow of information from DAN to protein in all eukaryotes[7]. Notably, the expression of spliceosome proteins in our study is persistently decreased throughout development in all 10 organs from infancy to adulthood (Fig. 2h, i). Q9CWZ3 (gene name *Rbm8a*) plays a critical role in the development of inhibitory neurons and *Rbm8a* perturbation leads to profound cortical deficits[36]. Protein P26369, also known as *U2af2*, can regulate Wilms' tumor gene 1 (WT1) to jointly play an important role in pre-mRNA splicing[37]. WT1 is crucial for normal kidney development, and mutations can cause kidney tumors and lead to

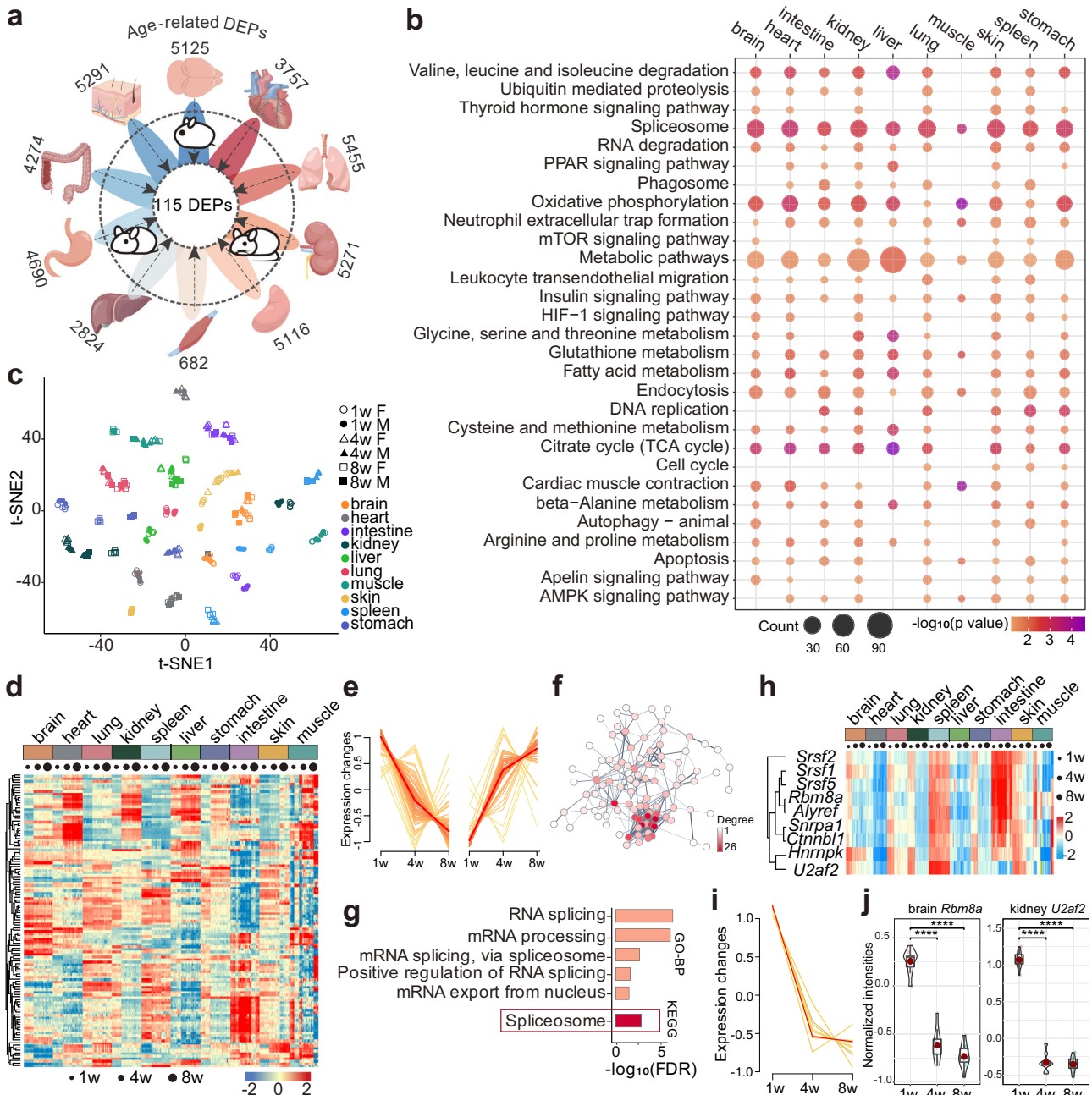

**Fig. 2 | Age-related differential proteins across ten organs from infancy to adulthood. a** The outer circle number indicates age-related differentially expressed proteins (DEPs) in each organ (the ANOVA cutoff of DEPs was set at adjusted *p*-value < 0.05). 115 age-related DEPs were detected by cross-comparing DEPs between 10 organs. Images of organs were generated by Figdraw, from https://www.figdraw.com. **b** Enrichment of KEGG pathway subsets of age-related DEPs in different organs. The color represents the raw -log$_{10}$ (*p*-value) and the size of the circle represents the number of proteins contained in the annotation. *p*-value was estimated in DAVID software using Fisher's Exact test. **c** t-distributed stochastic neighbor embedding (t-SNE) visualization of the 115 DEPs to cluster organs and ages. **d** Heatmap of the 115 DEPs expressed across different organs and ages. The heatmap is based on the abundance intensities of the significant DEPs normalized after unsupervised hierarchical clustering. **e** MFuzz analysis showed that 68 out of 115 DEPs were upregulated during 10 organs development, while 47 DEPs were downregulated during organ development. **f** Protein interaction network of the 115 DEPs. The network was visualized in Cytoscape and node colors are represented by Degree values. **g** GO biological process and KEGG enrichment analysis results of the 115 DEPs. **h** Organ-wise expression changes with age (within each column, from left to right) for the 9 spliceosome proteins in 10 organs co-expressing age-related DEPs. **i** MFuzz analysis showed that all 9 DEPs were downregulated during 10 organs development. **j** Dynamic expression changes of spliceosome protein *Rbm8a* in brain (left, *n* = 9 biologically independent samples) and *U2af2* (right, *n* = 10 biologically independent samples). Statistical analysis used two-tailed unpaired *t*-test with Holm-Sidak adjusted; exact *p*-value is indicated for levels of significance *p*-value < 0.0001(****). In the box plot, the median is represented by the center line and the box boundaries represent the first and third quartiles. Data are presented as mean values +/− SD. Source data are provided as a Source Data file.

Denys Drash or Frasier syndromes[38]. In our study, expression of *Rbm8a* in the brain and *U2af2* in the kidney decreased significantly from infancy to adulthood (Fig. 2j). Western blot experiments confirmed the trend of decreasing expression of the splice variants with age, which is consistent with the mass spectrometry experiment (Supplementary Fig. 5). All original images can be viewed from ProteomeXchange.

Based on the clustering of the samples, muscle samples were more discrete (Supplementary Fig. 6a). To avoid analysis bias caused

by discrete muscle proteins, resulting in a narrow co-expression range, we additionally analyzed nine organ co-expression DEPs without muscle samples. A total of 566 age-related DEPs co-expressed in 9 organs were detected (Supplementary Data 9). The t-SNE visualization of the DEPs data in 9 organs showed that all samples were aggregated by age and organ (Supplementary Fig. 6b). These 566 PGs were differentially expressed in different organs, but all exhibited age differences (Supplementary Fig. 6c). The enrichment analysis results of these 566 DEPs were equally enriched in processes such as RNA metabolism, which is similar to the enrichment results for 10 organ co-expression DEPs analysis (Supplementary Fig. 6d). The distinction is that among 9 organs, more signaling pathways for metabolic processes, including carbon metabolism, glutathione metabolism and amino acid metabolic pathways, were also found in the KEGG analysis of co-expressed DEPs.

Variations in the dynamic expression of proteins involved in development-related processes such as insulin signaling and glutathione metabolism were presented (Supplementary Fig. 6e, f). For example, the expression of glutathione metabolism protein tends to decrease with organ development in the spleen and intestine, and its expression increases in the brain, heart, kidney, liver and skin. Such results indicate the spatial and temporal differences in glutathione at the protein expression level during organ development. Notably, the trend of continuous decrease of spliceosome proteins in nine organs along with developmental progression is not spatially distinct between organs, which could serve as a potential universal biomarker for organismal development and maturation.

## Age-related DEPs among organs with high Pearson correlation coefficient

In the Pearson correlation analysis (Fig. 3a), we found common enrichment of proteins in several different tissue types such as lung, spleen, stomach, and skin. After the four organs—lung, spleen, stomach and skin—were independently filtered for valid values, a total of 6403 four organs co-expressed proteins were detected while 6165, 5900 and 5764 co-expressed PGs at 1-week, 4-week and 8-week, respectively (Fig. 3b, Supplementary Data 10).

To gain insights into the temporal behavior of the dynamic changes of proteins during the development of the four organs, ANOVA was used to analyze the age-related DEPs. Our results revealed a total of 3653 DEPs associated with age among the 6403 four organs co-expressed PGs in the four organs from infancy to adulthood (adjusted $p$-value < 0.05) (Fig. 3c). Unsupervised k-means clustering analysis of the 3653 DEPs resulted in two modules, named Module 1 and Module 2 (Fig. 3c). The module 1 contained 1452 PGs, which are mainly involved in pathways including mTOR signaling pathway, fatty acid metabolism and HIF-1 pathway (Fig. 3d). The expression of proteins in this module was dramatically raised with organ development (Fig. 3f). Apart from that, the module 2 contained 2201 PGs, which mainly participate in the spliceosome, DNA replication and cell cycle pathways (Fig. 3e). During organ development, protein expression of module 2 significant reduced (Fig. 3g). Numerous pathways, including mTOR signaling pathway, HIF signaling pathway, Notch signaling pathway and spliceosome, have been implicated in the regulation of organ development[39–42]. The synchronization of organ development is evidenced by the analogous expression trends of proteins involved in the regulatory pathways of the four organs, namely lung, spleen, stomach and skin. The results of organ developmental synchrony may be further explored by the concepts of "organ-organ cross-talk"[43,44] and "organ axes"[45].

DEPs are also enriched in many other pathways during development, such as DNA replication[46] and cell cycle[47]. The corresponding expression trends of proteins in these pathways were not consistent in the four organs (Supplementary Fig. 7). For instance, the spleen had a remarkable decline in cell cycle proteins while the other three organs

did not. The expression of proteins involved in oxidative phosphorylation in the spleen was not noteworthy increased with organ development. These results are consistent with the existing knowledge that inhibition of oxidative phosphorylation is beneficial to immune function[48]. The stomach is generally not reported to be metabolically active[12], whereas our investigations revealed that during development, the expression of proteins associated with fat acid metabolism and alanine, aspartate and glutamate metabolism were dramatically elevated in the stomach. Our findings not only unveiled specific proteins that are expressed similarly in the four organs, thus explaining how the four tissues cluster together during development, but also illustrated that functionally dissimilar organs may have similar molecular expressions during development.

In addition to the individual aggregation of the four organs mentioned above, functionally specific organ aggregation can also be found in the proteomic results of the kidney-heart as well as kidney-liver (Supplementary Fig. 8a, Supplementary Data 11). The detection of a large number of identical proteins between different organs at the same age was the main reason for organ aggregation (Supplementary Fig. 8b, c). 1820 age-related DEPs were found in the heart-kidney aggregation, while 2901 age-associated DEPs were found in the liver-kidney aggregation (Supplementary Fig. 8d, e). These proteins could be divided into two categories, one with upregulated expression along with organ development and the other was opposite. The enrichment results of these age-related DEPs co-expressed among different organs showed a consistent pattern, with the up-regulated proteins clustered in some metabolic pathways and the down-regulated proteins clustered mainly in the regulation of shear bodies (Supplementary Fig. 8f, g). These results re-emphasize the fact that spliceosome pathways and metabolic pathways, especially oxidative phosphorylation, carbon metabolism and fat acid metabolism pathways, are essential regulatory sway in the early-life development and maturation of different organs.

## Organ-unique age-related DEPs

Systemic diseases such as cancer, inflammation and developmental delays are still under-researched. Age factors have been commonly noted in systemic diseases. In our study, we analyzed the correlation of proteomic data in 10 organs at 1-week, 4-week and 8-week respectively. The results shown that the correlation between organs decreases from 0.80 at 1-week, to 0.72 at 4-week, and further to 0.69 at 8-week (Supplementary Data 12) during the period from infancy to adulthood as each organ develops and matures functionally (Supplementary Fig. 9). When we analyzed developmentally related DEPs in individual organs, there were age-related organ-unique DEPs that were intimately associated with development in addition to the proteins that were co-expressed between the organs as indicated above (Supplementary Data 13). Among the DEPs unique to the organ, brain-unique age-related DEPs had the largest number at 675, followed by lungs, spleen, intestine, skin, kidney, liver and stomach (Fig. 4a). Heart and muscle-unique DEPs counted below 100. As shown in Fig. 4b, the top five pathways for organ-unique age-related DEPs were mainly engaged in organ-specific functions. For instance, kidney-unique age-related DEPs participated in urate transport, carbohydrate transport and renal system process. This is consistent with the impetus definition of the kidney as a functional organ.

The brain is one of the most complex organs in vertebrates. The analysis of brain-unique proteins revealed that two modules of age-related DEPs were detected. The module 1 showed an up-regulation in protein expression with organ development, whereas module 2 showed the opposite pattern (Fig. 4c, Supplementary Data 14). Functional enrichment analysis indicated that pathways associated with the neuronal system, behavior and glutamatergic synapse in module 1 (Fig. 4d, Supplementary Data 14). Meanwhile, module 2 proteins were engaged in the neuron projection development, brain

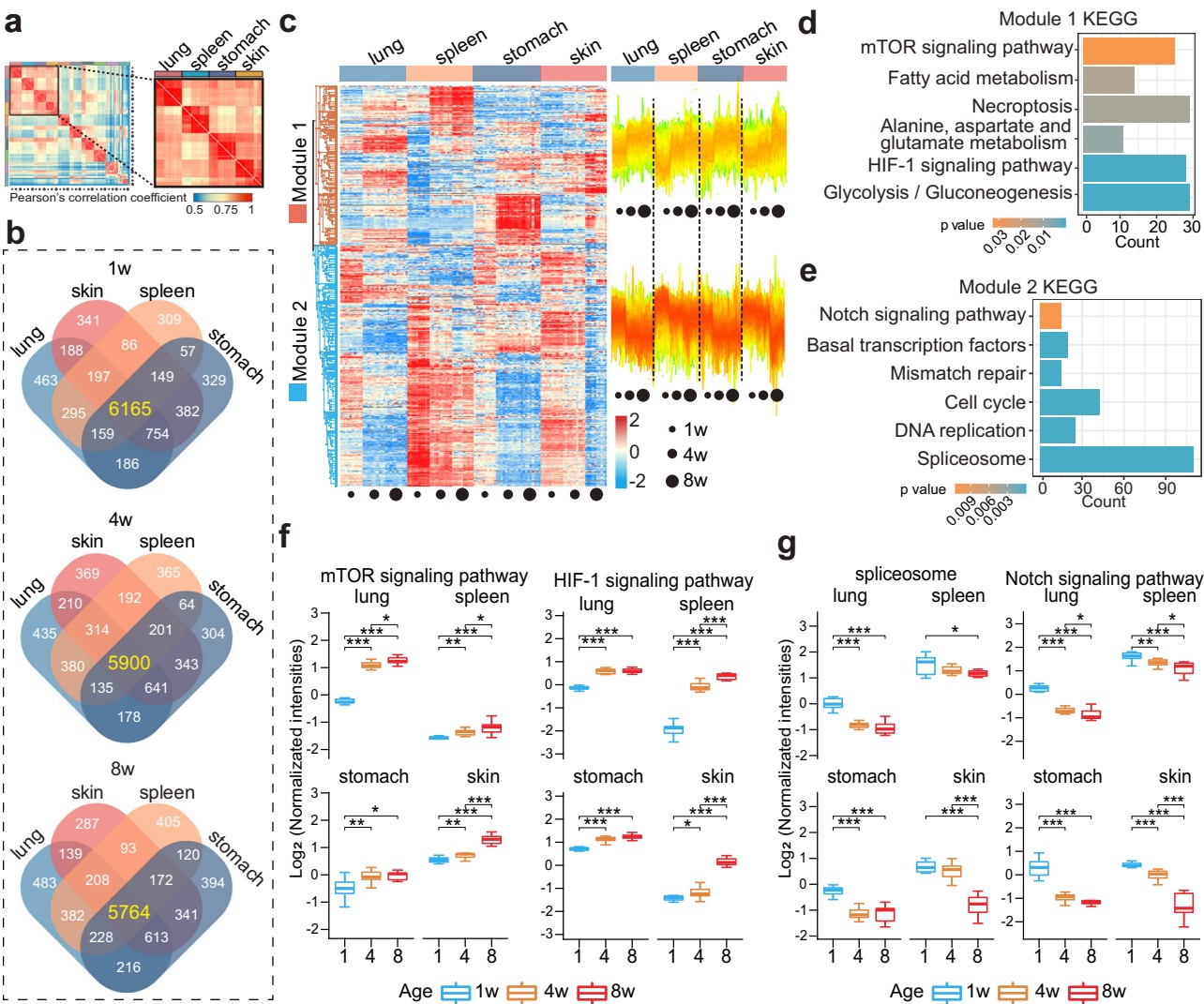

**Fig. 3 | Expression of age-related DEPs in lung, spleen, stomach and skin. a** The four organs, namely lung, spleen, stomach and skin are clustered together in Fig. 1 d, indicating that they are more proteomic functionally similar to each other. **b** Venn diagram of protein expressions between lung, skin, spleen, and stomach from infancy to adulthood. Top: 1-week; Middle: 4-week; Bottom: 8-week. **c** Heatmap of the two expression modules revealed by k-means clustering in the lung, spleen, stomach, and skin (left). The protein expression trends can be divided into two modules, denoted module 1 (a class of proteins whose expression is upregulated from infancy to adulthood) and module 2 (a class of proteins whose expression decreases from infancy to adulthood). KEGG analysis for module 1 with

up-regulated proteins (**d**) and module 2 with down-regulated proteins (**e**). *p*-value was estimated in DAVID software using Fisher's Exact test. Temporal expression patterns of proteins involved in the core KEGG pathways of up-regulated proteins (**f**) and down-regulated proteins (**g**). *n* = 10 biologically independent samples. Statistical analysis used two-tailed unpaired *t*-test; exact *p*-value are indicated for levels of significance *p*-value < 0.05(*), *p*-value < 0.01(**), *p*-value < 0.001(***), *p*-value < 0.0001(****). In the box plot, the median is represented by the center line and the box boundaries represent the first and third quartiles. Data are presented as mean values +/− SD. Source data are provided as a Source Data file.

development and cellular response to calcium ions were enriched. Huntington's protein O35668 (gene name *Hap1*) decreased from 1-week to 4-week and remained stable until adulthood (Fig. 4e). *Hap1* was originally detected as a neuronal protein that was specifically associated with huntingtin (Huntington's disease)[49]. The link between *Hap1* and huntingtin may be eliminated by downregulation in Hap1 abundance, causing the release of toxic mutant huntingtin. Besides, Hap1 is critical for postnatal development, since *Hap1*-KO mice frequently die before P3 due to inhibited food intake behavior[50,51]. Moreover, the expression of some proteins that are closely related to brain development were also upregulated with brain development. Q01097 (gene name *Grin2b*) is an important component of N-methyl-D-aspartate (NMDA) receptor complexes and regulates synaptic plasticity by binding D-serine and coordinating trans-synaptic complexes[52]. The expression level of *Grin2b* increased and remained stable after 1-week (Fig. 4f), implying the progression of long-term

neurodevelopment rapid growth between 1 and 4-week and remains stable after 4-week.

The instrument we used in this study was an EASY nLC 1200 coupled with QE HF-X system (Thermo Fisher Scientific). We further validated our brain-unique DEPs results on an additional timsTOF flex (Bruker) mass spectrometry platform. The results showed that both mass spectrometry detected 6671 PGs in the brain, about 78% of the total amount (Supplementary Fig. 10a, Supplementary Data 15). Heatmaps and expression trends of the two mass spectrometry platforms regarding brain-unique age-related DEPs (ANOVA adjust *p*-value < 0.05) were shown in Supplementary Fig. 10b. The number of age-related down- and up-regulated DEPs detected by both platforms was 221 and 343, respectively (Supplementary Fig. 10c). PCA analysis also showed the consistency of the mass spectrometry data generated by the two different platforms (Supplementary Fig. 10d). Moreover, we generated quantile-quantile (Q–Q) plots[53] to compare the *P* values

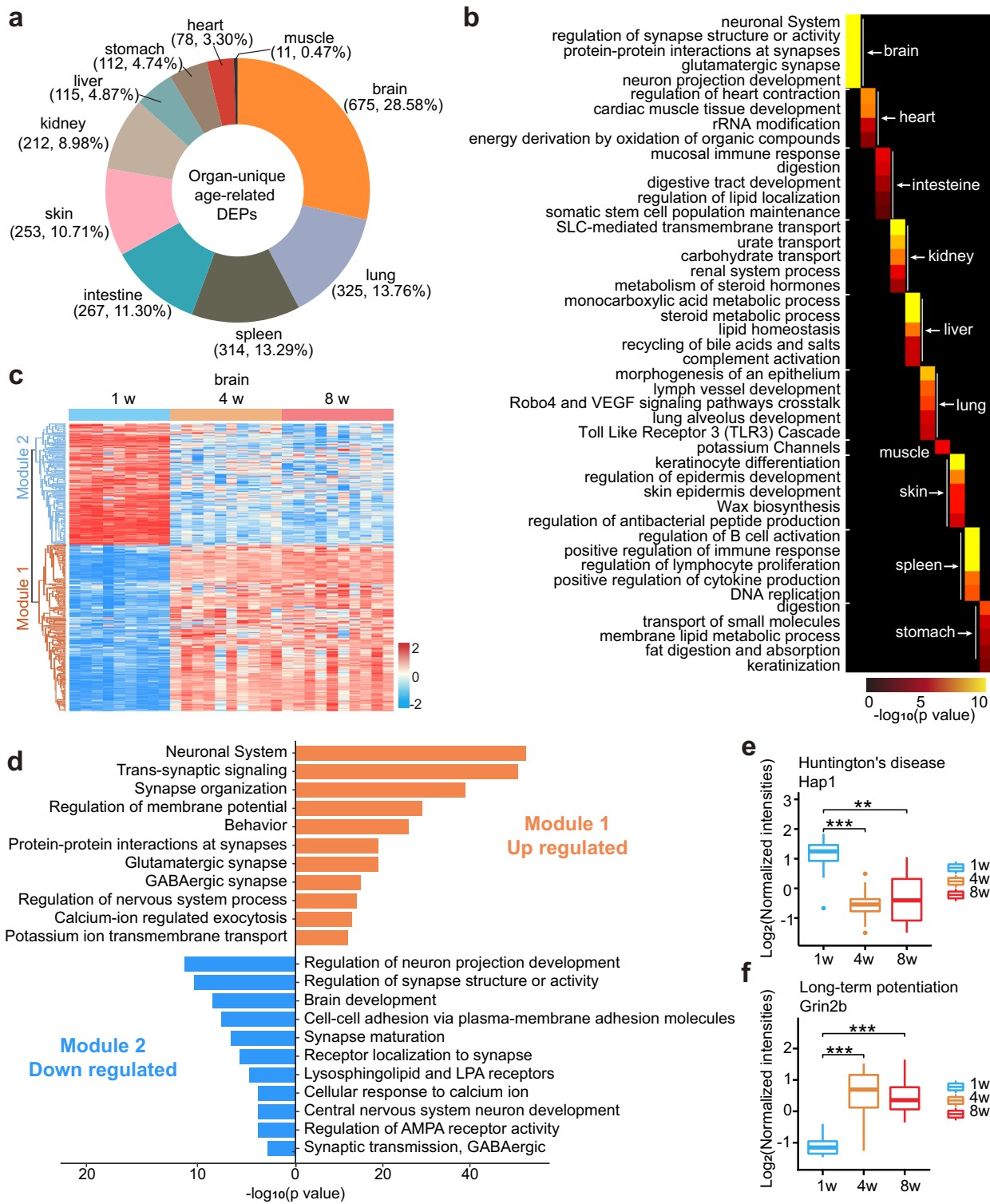

**Fig. 4 | Organ-unique age-related DEPs. a** Number of age-related DEPs unique to every single organ. Among the 10 organs, the brain shows the largest organ-unique protein subset, while muscle shows the smallest organ-unique protein subset. **b** The top five non-redundant enrichment pathways from each organ-unique age-related DEPs group. **c** Heatmap of brain-unique age-related proteins by semi-supervised hierarchical clustering. The protein expression trends fall into two modules, with proteins in module 1 being upregulated with organ development and those in module 2 being downregulated with organ development. **d** Enrichment analysis of DEPs from two modules in **c**. *p*-value in **b** and **d** were estimated in Metascape

software. Temporal expression patterns of proteins involved in Huntington's disease (**e**) and long-term potentiation (**f**). *n* = 9 biologically independent samples in 1-week brain group, *n* = 10 biologically independent samples in all other groups. Statistical analysis used two-tailed unpaired *t*-test; exact *p*-value is indicated for levels of significance *p*-value < 0.01(\*\*), *p*-value < 0.001(\*\*\*). In the box plot, the median is represented by the center line and the box boundaries represent the first and third quartiles. Data are presented as mean values +/− SD. Source data are provided as a Source Data file.

reported of the data detected by the two mass spectrometers with the normalized rank *P* values. that represent the uniform distribution to illustrate the consistency of the data obtained by the two instruments (Supplementary Fig. 10e). The data points exhibit an evenly distributed pattern along the diagonal, indicating strong reproducibility of the detected data across both platforms.

Top non-redundant enrichment results for organ-unique age-related DEPs were primarily engaged in core organ functions (Supplementary Fig. 11, Supplementary Data 13). For example, lung alveolus development and endothelium development were found in the lungs, regulation of lymphocyte proliferation and myeloid cell development were found in the spleen, and skin epidermis development and regulation of antibacterial peptide production were found in the skin. Of note, some immune-related processes were found in the enrichment results in organ-unique proteins, such as the TLR pathway, lymphatic vessel development and mucosal immune responses. This implies that immune processes widely participate in organismal organ development and function formation.

## Age-unique proteins in ten organs

Protein expression in organs is highly dynamic as the organs develop. We have found that some proteins showed temporal patterns and only expressed at a specific time. To detect these age-unique proteins, we screened for proteins acquired at different ages in each organ (Fig. 5a, Supplement Data 16). In addition to muscle, the number of proteins that were expressed from infancy to adulthood accounted for more than 75% of the proteins detected in this organ at all time points (Supplementary Fig. 12a). The proportional stacking diagram visualized the proportion of age-unique proteins in each organ (Fig. 5b). Except for the intestine and skin, the other organs showed the highest amount of age-unique proteins at 1-week.

When comparing the commonalities of age-unique proteins in all organs, only P09535 (Insulin-like growth factor II, gene name *Igf2*) expression was found in all 10 organs at 1-week (Fig. 5c). *Igf2* expresses multiple transcripts in many tissues during the embryonic and neonatal stages as a major fetal growth hormone in mammals and plays a key role in regulating fetoplacental development, while its expression in adult animals is limited to the choroid plexus and the light meninges[54,55]. However, *Igf2* was widely expressed in the vital organs of the 1-week mice in this study, but not in the 4-week or 8-week mice, demonstrating that *Igf2* is not only essential for fetal growth but perhaps continues to function in postnatal development. At 4 and 8-week, few age-unique proteins were co-expressed between organs. Only 3 age-unique proteins appeared in 2 organs at 4-week (Supplementary Fig. 12b). At 8-week, the highest number of age-unique inter-organ expressions was 18, which appeared in the intestine and skin (Supplementary Fig. 12c).

To further investigate the function of age-unique proteins, we used DAVID to conduct an enrichment analysis of age-unique proteins in corresponding time points (Supplement Data 16). Our results showed that 1-week-unique proteins were mainly involved in organ development processes such as cell cycle, protein phosphorylation, cell division and RAN splicing (Fig. 5d). Nevertheless, 4-week unique proteins participated in immune response, lipid system process and tissue development process (Fig. 5e). 8-week unique proteins engaged in phosphorylation, ion transport and immune response (Fig. 5f).

Particularly, the 8-week unique proteins had a larger proportion of age-unique proteins in the intestine and skin than in other organs. (Fig. 5b). Although the skin and intestine have different physiological structures, they share many similar characteristics as surface organs exposed to the external environment. In our study, 1-week-unique proteins both in the intestine (Supplementary Fig. 12d) and skin (Supplementary Fig. 12e) mainly participate in a developmental process such as cell division, hippo signaling and WNT signaling. WNT[56] and the hippo signaling pathway[57] are well-known as important

regulatory pathways for organism development and fate choices during organ morphogenesis. However, 4-week and 8-week unique proteins in the skin and 8-week unique proteins in the intestine were shown in the immune process including defense response to Gram-positive bacterium, neutrophil degranulation, NF-kappa B signaling and IL-2 signaling pathway. The specific expression of immunomodulatory proteins at 4 and 8-week may be the result of changes in diet and constant exposure to external environmental microorganisms during the development of the skin in early life. This means that when investigating skin development, environmental factors shall be considered.

## Sexually dimorphic proteins in organ development

Recent reports demonstrate a male bias in most cancers and a female bias in longevity[58,59]. Sexual dimorphism in organ-developing proteins remains intricate. To gain further insight into the sex influence during organ development and maturation, the Student's *t*-test was performed to analyze the sex-related DEPs changes between males and females in 1-week infancy mice (Fig. 6a), 4-week young mice (Fig. 6b) and 8-week adulthood mice (Fig. 6c, Supplement Data 17). When comparing sex-related DEPs of organs at the same age, we found that the liver co-expressed the highest number of sex-related DEPs with other organs (Supplementary Fig. 13a–c). At 1-week, the muscle and liver had the highest number of differentially expressed PGs (435). At 4-week, the liver and intestine had the highest number of differential PGs (219). At 8-week, the number of DEPs co-expressed in skin and liver was the highest (304). Sex-related DEPs expressed in both organs were involved in basic life processes such as mRNA processing, translation and intracellular protein transport (Supplement Data 17).

In our study, some sex-related DEPs were expressed from infancy to adulthood, while other proteins appeared only at a specific age. As shown in Fig. 6d, we detected the largest number of sex-related DEPs in the liver (4659 in total). Furthermore, the number of sex-related DEPs in the liver was similar at 1, 4, and 8-week, and the number of proteins that showed sex differences from infancy to adulthood was the highest in all 10 organs (Supplementary Fig. 13d). This suggests that sex differences during the early-life development of the liver are the most remarkable among all 10 organs. Moreover, only the liver, spleen, heart, kidney, lung and muscle were found to have sex-related DEPs at all three developmental stages. There were 554 sex-related DEPs in the liver throughout 1, 4 and 8-week of development, followed by the spleen with 127 (Supplementary Fig. 13d). The number of sex-related DEPs present in heart, muscle and lung at all three ages was less than 15. These results suggest that the quantity and variety of sexual dimorphic proteins vary among organs at different stages of organismal development. PCA analysis of DEPs across age in the liver shows clearly identifiable sex boundaries (Fig. 6e). The enrichment results showed that the proteins with decreased expression in males compared to females were mainly involved in retinol metabolism, nucleocytoplasmic transport, endocytosis, metabolic pathways (Fig. 6f, top). And the up-regulated proteins in male than female expression were mainly involved in RNA degradation, mRNA surveillance pathway and spliceosome pathway (Fig. 6f, bottom). Protein-protein interactions analysis found 23 spliceosome proteins in liver sex-related DEPs (Fig. 6g). 5 spliceosome proteins were highly expressed in female liver, and another 18 spliceosome proteins were highly expressed in male livers (Fig. 6h). *Snrpd3* plays a role in pre-mRNA splicing as a core component of the spliceosomes U1, U2, U4 and U5 small nuclear ribonucleoproteins, and it is significantly more expressed in female livers than in males (Supplementary Fig. 13e, top). The removal of plasma LDL by the liver is effective in reducing plasma protein sterol levels and thus the risk of atherosclerotic cardiovascular diseases[60]. U2-spliceosome-associated gene *Rbm25* is related to low-density lipoprotein cholesterol levels in humans[61]. *Rbm25* shows significant liver sex dimorphic in early development (Supplementary Fig. 13e, bottom). This suggests

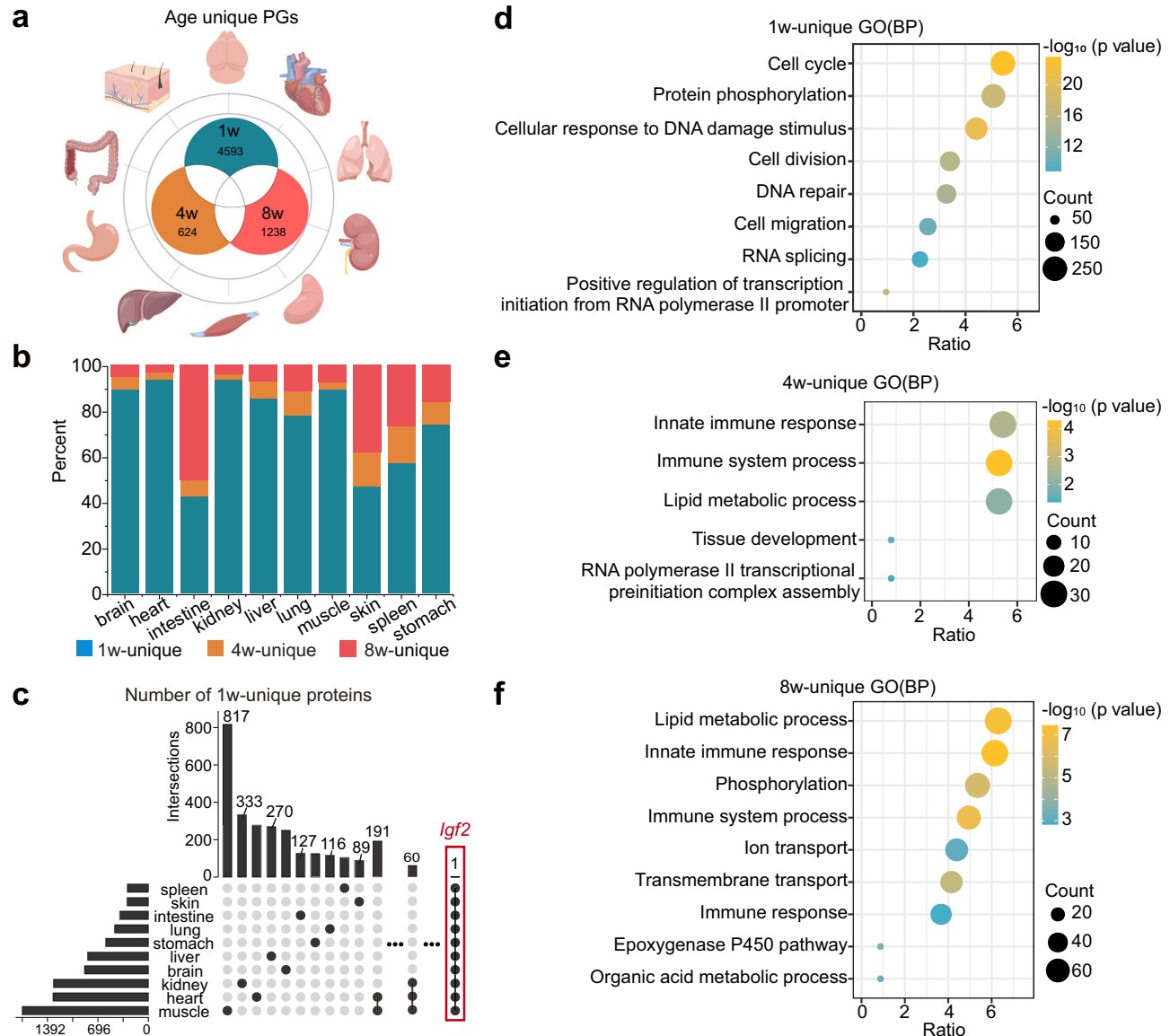

**Fig. 5 | Age-unique proteins in ten organs. a** Proteins detected at different ages were compared to obtain age-unique proteins. Images of organs were generated by Figdraw, from https://www.figdraw.com. **b** Proportion stack bar chart of age-unique protein groups in each organ. Except for the intestine, the proportion of 1-week unique proteins was the highest among the 9 organs. **c** Number of 1-week unique proteins in 10 organs. The number of proteins in each organ that are only expressed in 1-week organ (bars, left). Protein counts (top, bars) are found in each combination of organs or a single organ (bottom, black dots). Only 1 of the 1-week unique protein, P09535 (Insulin-like growth factor II, *Igf2*), was detected in all 10 organs at 1-week. Gene Ontology (GO) biological process enrichment analysis ensembles a total of 4593 1-week-unique proteins. **d** 624 4-week-unique proteins (**e**), and 1238 8-week-unique proteins (**f**) from all 10 organs. The size of the circle represents the number of proteins contained in the annotation and the color represents the raw -log$_{10}$ (*p*-value). *p*-value was estimated in DAVID software using Fisher's Exact test. Source data are provided as a Source Data file.

that we may be able to study the sex-related aspects of atherosclerotic cardiovascular disease from the sexual dimorphic of Rbm25 gene and protein.

## Discussion

Early-life development is a multi-organ interactive process involving the regulation of thousands of molecules. A slight difference occurs in early life could induce huge dividends in later phenotype[3,62]. This organism-wide characterization of early-life dynamic proteomics atlas fills in the knowledge gap for early-life multi-organ development and maturation.

Systematic and simultaneous detection of proteomic data from multiple organs under the same conditions could provide a more comprehensive picture of in vivo early life development. Existing mass

spectrometry-based quantitative proteomics for organ development studies mostly focus on single organs. However, the available mass spectrometry data demonstrated inconsistent reproducibility of protein data performed between different laboratories and at different times[17]. Our study is strictly controlled by the selection of animal samples, sample pre-treatment methods, mass spectrometry acquisition strategies, and protein detection and quantification methods. Precisely for this reason, our study depicts a time-lapse proteomic shift in the development of 10 major organs from infancy to adulthood mice with a coverage of 11,533 PGs. Such a dataset complements the unmet need by single organ studies. The DEPs detected in our study have been validated for reproducibility between at least 5 biological replicate samples with guaranteed reliability. Additionally, our main results have been validated through WB experiments and different mass

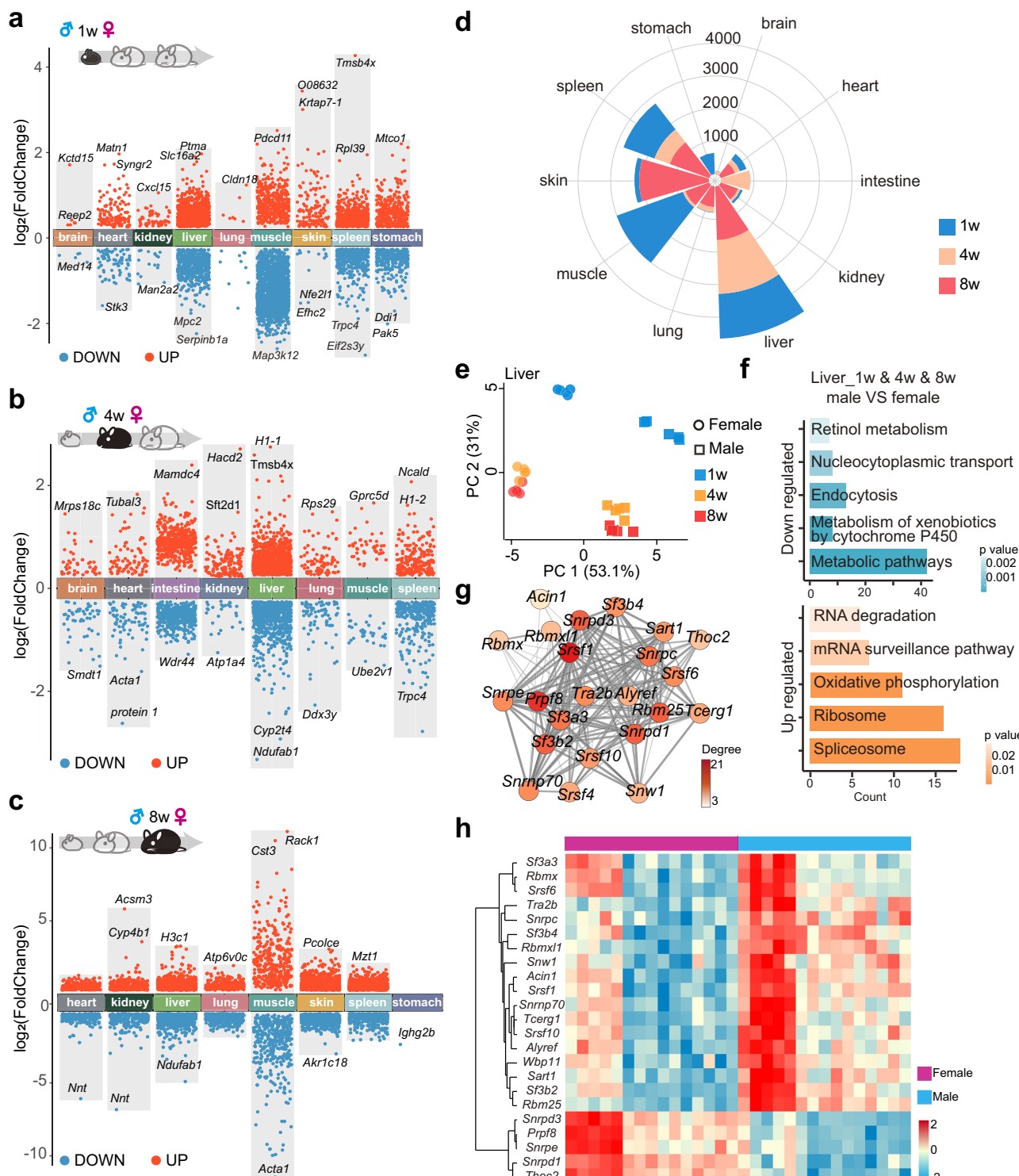

**Fig. 6 | Sexually dimorphic protein expression during organ development.**
1-week (**a**), 4-week (**b**) and 8-week (**c**) sex-related DEPs indicating up- and down-regulated expressions across organs. $n = 9$ biologically independent samples in 1w, $n = 10$ biologically independent samples in 4w and 8w respectively. Statistical analysis used two-tailed unpaired $t$-test. An adjusted $p$-value < 0.05 and fold change > 1.2 is set as the cutoff for up-regulated (red) proteins, while fold change < −1.2 is set as the cutoff for down-regulated (blue) proteins. **d** Distribution of the number of sexual dimorphic proteins quantified across ten organs at three ages. **e** PCA clustering of 554 liver sexual DEPs across 3 ages. **f** KEGG enrichment clusters from sexual DEPs in the liver at all 3 ages. The top is the pathways that are downregulated

in males than females. The bottom is the pathways that are upregulated than females. The bar graph length represents the number of proteins contained in the annotation and the color represents the raw $p$-value. $p$-value was estimated in DAVID software using Fisher's Exact test. **g** Protein interaction network of sex-differentiated spliceosome proteins appear in the liver at 1-, 4- and 8-week. The network was visualized in Cytoscape and a discrete color scale is used to represent the betweenness value. **h** Heat map analysis was performed by semi-supervised hierarchical clustering of 23 sexual dimorphic spliceosome proteins that were present in the liver from infancy to adulthood. Source data are provided as a Source Data file.

spectrometry platform. The data from the combined assay of multiple organs from infancy to adulthood would allow us to paint a sweeping picture of the early-life development of the mouse whilst revealing the various identical and variable biochemical and physiological processes involved in the development of the living organism as an organic whole.

Regulation of proteins fundamental to the developmental fate of organisms can vastly improve the index of health. In our study, differential protein expression from infancy to adulthood showed that 115 PGs were profoundly engaged in the development of all 10 organs. The biological functions of many proteins depend on specific physical interactions with other proteins, and defining a complete map of protein-protein functional interactions in an organism has long been regarded as a pivotal objective in the post-genomic area[63,64]. String analysis showed that 96 out of 115 DEPs could form a protein interactions network, implying that these inter-organs age-related DEPs could be facilitated to defining the complete map of functional protein-protein interactions and certainly plausible results need to be verified by more detailed experiments. The enrichment results of the 115 DEPs indicated that these proteins were mainly enriched in RNA and mRNA splicing process. Similar to our study, developmental studies of existing single-organ such as the liver[11,14], stomach[12], lung[13], brain[65] and intestine[15] also showed that the differential proteins and genes were participated in the splicing process and spliceosome pathways. Thousands of protein isoforms are derived from splicing changes, and these isoforms are linked to crucial physiological processes such as cell differentiation, organ development, and tissue homeostasis[66]. By comparing the different protein isoforms identified at different stages, differences in splicing events might be inferred. Further research might be conducted by optimizing the detection accuracy of proteomics or utilizing transcriptomics and protein sequencing to address the relevant issues[7,67].

This study has also proposed a set of identity-acquiring organ-unique age-related DEPs that are crucial to the specific function of the corresponding organs. These findings can contribute to a better parsing of the independent development of single-organ and the overall development of organisms. By making further connections, these findings could bridge the gap between individual organs and the organic whole, with important implications for the study of organ organization, organ-specific and systemic diseases. Interestingly, the inter-age correlation of the skin is different from other organs, and our proteomics data show that 4-week-old skin is closer to 1-week-old skin. This result is similar to that of Dekoninck et al. in murine tail and paw epidermis[35], both of which indicate that mouse skin undergoes rapid development until 4 weeks, after which it enters a period of maturation and stabilization. All of these phenomena pointed to commonalities in organ development as well as many organ-unique characteristics.

The 3 time points adopted in this study may not well cover the entire cycle of early life. The aggregation of protein data at 4 and 8-week indicates that the protein expression profile at 4-week is approximates that of adult mice, suggesting that more time points before 4-week are desirable. However, the comparison between 1 and 8-week could also indicate changes in protein expression in various organs during early life. The protein profile at 4-week could reflect the changes in the organism after weaning.

Although we use the DIA method to reduce the stochasticity of mass spectrometry detection, low-abundance but critical proteins may not be detected due to the limitations of current mass spectrometry detection accuracy and the difficulty of complex data resolution by DIA. Despite these limitations, we believe our dataset provides a valuable scientific resource for multidisciplinary research domains including but not limited to early-life organism development, aging-related diseases, systemic diseases and shaping a healthy organism.

## Methods

### Animals and organ collection

All experimental procedures were approved by the Institutional Biomedical Research Ethics Committee at the Institutional Animal Care and Use Committee (IACUC) of Shanghai Jiao Tong University, Shanghai, China. C57BL/6 J mice were ordered from Shanghai Jiesijie animal Co., Ltd. Mice were randomly assigned to different age groups of 4–5 mice per cage according to sex and housed in a light-dark cycle of 12:12-hr, the temperature of 20–26 °C, the humidity of 40–60%, and provided with water and standard rodent chow. Mice were deeply anesthetized using sodium pentobarbital (50 mg/kg, intraperitoneal injection, Sigma-Aldrich) then perfused with 20 ml of pre-cooled 1× PBS via the myocardium and dissected. First, the intact organs were collected including heart, lung, liver, spleen, kidney, and brain and placed in EP tubes. Second, the stomach and intestine were rinsed with ice-cold PBS after removing the internal digestive material, dried, and placed in EP tubes. Then the calf gastrocnemius muscle was extracted as a muscle sample. Finally, 1 cm² of skin tissue was removed from the neck of mice and placed in a tube after fat removal. 4- and 8-week old mice were first hairless and 1-week mice were not. All samples were placed in liquid nitrogen as soon as possible after stripping and stored at −80 °C until protein extraction.

### Homogenization and digestion

There are in total 300 samples from 10 organs from 30 individual mice. 5 females and 5 males at each time point. The sample preparation method was as described Filter-aided sample preparation (FASP)[31]. The detailed descriptions were as follows:

About 10–20 mg of tissue were weighed and rapidly frozen in liquid nitrogen, then ground in a mill at 60 Hz and −20 °C for 40 seconds. Next, lysate containing protease inhibitor was added to tissue at a concentration of 20 μl/mg before grinding again. After sonication on ice, the remaining debris was removed by centrifugation at 10,000 g for 20 min and the supernatant lysates were quantified by BCA assay (Thermo Fisher Scientific).

For digestion, a total of 100 μg protein per sample was precipitated in acetone overnight at −20 °C. The protein pellet was obtained after centrifugation at 14,000 g for 20 min, and the pellet was washed twice with acetone. After discarding the supernatant and evaporating the residual acetone, the pellet was resuspended in 7 M guanidine hydrochloride. The protein solution was reduced with 10 mM dithiothreitol (DTT) at 55 °C for 1 hour and alkylated by 20 mM iodoacetamide (IAA) for 30 min in the dark. Samples were washed with ammonium bicarbonate on 10 kDa ultrafiltration concentrators (Sartorius) to purify the protein. Finally, the trypsin (Promega) digestion was performed overnight at 37 °C using a 50:1 protein: protease ratio (w/w). After termination of digestion by the addition of a final concentration of 1% formic acid (FA), the peptides were desalted using a C18 spin column (Monospin) according to the manufacturer's instructions. Peptides eluted were dried under vacuum and stored at −80 °C until subsequent use for liquid chromatography tandem mass spectrometry (LC-MS/MS) analysis.

### LC-MS/MS Analysis

MS analysis was performed on an EASY nLC1200 coupled with QE HF-X system (Thermo Fisher Scientific). The samples were resuspended in 0.1% FA, and adjusted to the final concentration to 0.5 μg/μl. After spiked with iRT (Biognosys), 0.5 μg of each sample was loaded onto 2 cm × 75 μm homemade reversed-phase C18 column (particle size, 3 μm; pore size, 100 Å. Thermo Fisher Scientific) and separated by 25 cm × 75 μm analytical column (particle size, 2 μm; pore size, 100 Å. Thermo Fisher Scientific). Separation was performed over a 120 min gradient at a flow rate of 300 nl/min. 0–1 min with buffer B ranging from 1% to 8%; 1–81 min with buffer B ranging from 8% to 22%; 81–98 min with buffer B ranging from 22% to 28%; 98–112 min with

buffer B ranging from 28% to 36%; 112–116 min with buffer B ranging from 36% to 100%; 116–120 min with buffer B 100%. The mobile phase A and B were 0.1% FA in water and 0.1% FA in 80% acetonitrile, respectively.

The DDA strategies were used for both quality control samples Hela cell digested and tissue mixed samples to assess instrument status after the mass spectrometer cleaning or calibration. The DDA strategies were set as below: MS1 at a resolution of 60,000, AGC target 3E6 charges. The top 20 precursors were subject to ddMS2 scan with an m/z range of 200–1200 with AGC target 2E5 charges at a resolution of 15,000, max IT was 25 ms, and normalized collision energy (NCE) was 28.

The DIA acquisition strategy consisted of an m/z range from 320 to 1220 at 60,000 resolution and AGC was setting 3E6. All precursors in a cycle transferred to MS/MS scan used an isolation width of 20 m/z, a set of 45 overlapping windows (1 m/z overlap), covering the mass range of 330–1210 m/z. Precursor ions with +1 and +6 or higher charge states were excluded from sequencing.

Trypsin digests of Hela cells as quality control samples were assayed before each organ cohort started and after routine equipment calibration to ensure instrument condition. The sample mixture was assayed as a control to ensure good sensitivity and reproducibility.

Mass spectrum validation was performed on the nanoElute coupled with Tims TOF flex (Bruker). Sample separation in a 15 cm × 75 μm, C18 1.7 μm column (AURORA ELITE). Peptide separation was performed on a 45 min gradient at a 300 nl/min flow rate. 0–25 min with buffer B ranging from 7% to 30%; 25–38 min with buffer B ranging from 30% to 58%; 38–45 min with buffer B ranging from 58% to 80% and held until 45 min. The mobile phase A and B were 0.1% FA in water and 0.1% FA in 100% acetonitrile, respectively. Acquisition schemes for the dia-PASEF methods[68] as below: mass width (25 Da), number of mobility windows (2), mass range (350 to 1200 m/z) and in both scan modes, the collision energy was ramped linearly as a function of the mobility from 59 eV at $1/K0 = 1.3$ V.s/cm$^2$ to 20 eV at $1/K0 = 0.75$ V.s/cm$^2$.

### Protein detection and quantification

Proteome Discoverer 2.4.0.305 (Thermo Fisher Scientific) was used for DDA raw data protein detection. The raw files were searched against the mouse proteome database (https://www.uniprot.org/uniprotkb?query=UP000000589) from the Universal Protein Resource (UniProt). The PD search strategy was as follows. At least one unique peptide with a minimum 6 amino acid length was required for protein detection. The maximum trypsin missed cleavages were 2. The false discovery rate (FDR) < 1% of the percolator was sited to filter peptides and peptide spectral matches and peptide. All DIA raw files were analyzed using the DIA-NN-1.8.1 (https://github.com/vdemichev/DiaNN)[21]. The DIA-NN workflow is briefly described as follows: the chromatograms for each precursor ion and its fragment ions are extracted. Putative elution peaks are then scored, and the highest-scoring peak is selected. Potentially interfering peptides are identified and eliminated. The matches between precursors and peaks enable the calculation of q-values using a group of DNNs, and interferences are also removed from the fragment elution curves. The DIA data search strategy followed the default parameters for the library-free analysis search method in the instructions. Specifically, N-terminal acetylation and methionine oxidation were set as variable modifications, whereas cysteine carbamidomethylation was set as a fixed modification. The trypsin missed cleavage was set to 1. Peptides range in length from 7 to 30, and precursor FDR was stetted 1.0%. All protein groups were identified based on a 1-peptide minimum and an FDR of 1.0%. Using a 1-peptide minimum although more reliance on confidence scoring systems to filter out false positives but can increase the sensitivity of protein identification and maximizes the use of the available data. One brain sample at 1-week was removed due to abnormal mass spectrogram acquisition, and 299 mass spectra were used for data analysis.

### Western blot analysis

Protein samples from tissue homogenates were mixed with 5× protein loading buffer and heated at 98 °C for 10 min. 20 μg protein lysates were then loaded each well and separated on 8% SDS-PAGE gel and transfer to a nitrocellulose membrane. After the membrane was blocked with 3% BSA (Bovine Serum Albumin) in TBST for 2 h, the membrane was incubated with primary and secondary antibodies according to the manufacturer's instructions. The nitrocellulose membranes were visualized by ECL.

### Bioinformatics analysis

Perseus software (v1.6.15.0), R (v4.2.0), Origin (v2022b), and "Wu Kong" platform were used for bioinformatic analysis. The identified PGs undergo rigorous filtering, and only those identified in at least four out of five biological samples are included for subsequent analysis. Coefficient of variance (CV) is used to assess the relative variability of the dataset[69] and median normalized abundance was calculated using data after log2-transformation, followed by missing value imputation. Group differentially expressed PGs were calculated by one-way ANOVA test with the Benjamin-Hochberg procedure under $p$-value < 0.05. For pairwise comparisons at different age points in each organ, data between two-time points which after missing values filter, imputation, and coefficient variation filter were analyzed using the Student's $t$-test (two-tailed) and only statistically significant proteins (log2(fold-change) cutoff >1.0; Benjamin-Hochberg adjusted $p < 0.05$) were included in the downstream analysis.

Gene ontology (GO), Kyoto Encyclopedia of Genes and Genomes (KEGG) terms functional enrichment analysis and annotation were performed using the web-based tool DAVID and Metascape (http://metascape.org.).

### Statistics and reproducibility

In addition to the sample sizes, exclusion criteria, and statistics discussed in the rest of the Methods section, more information can also be found in the Reporting summary. In this study, no statistical method was used to predetermine sample size and our study did not involve groups that necessitated blinding.

### Reporting summary

Further information on research design is available in the Nature Portfolio Reporting Summary linked to this article.

## Data availability

The mass spectrometry proteomics data and searching output data have been deposited to the ProteomeXchange Consortium via the PRIDE partner repository with the dataset PXD041400. Source data are provided with this paper.

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

## Acknowledgements

We thank Wenqiong Su for sample preparation; Rui Chen and Kaiyuan Li for data curation; Yiyang Li, Chunhui Zhai and Guangxia Shen for discussions on the experiment design. This work was supported by the National Key R&D Program of China (2022YFF0710200, 2022YFC2601700) and NSFC Projects (82361148715, T2122002, 22077079), Shanghai Municipal Science and Technology Project(22Z510202478), Shanghai Municipal Education Commission Project(21SG10). Thanks to AEMD SJTU, Shanghai Jiao Tong University Laboratory Animal Center for the support.

## Author contributions

X.D. and Q.W. designed and directed the study. Q.W. wrote the original draft. X.D. and L.J. edited the manuscript. Q.W., XW.D., Y.D., S.S., Y.C. and S.Z. performed the sample collection. Q.W., A.W., Z.X. and L.W. performed mass spectrometric measurements. Q.W., Z.X. and B.W. performed all data processing and statistics.

## Competing interests

The authors declare no competing interests.
