## [Peer Review File · Nature Communications]

Ten-organ proteome atlas from infancy to adulthood mice.REVIEWER COMMENTS

Reviewer #1 (Remarks to the Author):

In their manuscript entitled „Ten-organ developmental proteome atlas from infancy to adulthood mice“, Wang et al describe the results of a DIA-based proteomic study investigating the changes in proteomic composition in ten mouse tissues. Using a DIA approach the authors claim to have identified over 11.000 proteins and have identified candidate proteins that change with developmental stage. The presented dataset has high potential value for the community. While the rationale of the study is well defined, and methods, data and evaluations are (with very minor exceptions) well documented in the supplementary datasets, the manuscript suffers from a number of potential major issues.

Major points:

1. Data quality: some of the muscle samples (all 5 biological replicates of the 8 week timepoint , male) show only a very weak correlation to any other samples of the entire study, this might indicate a problem in sample preparation – this is in contrast to the statement in line 108 and will impact the number of significantly regulated proteins between male and female mice.
2. For most tissues (brain, heart, kidney, liver, lung) the correlation analyses indicate a high degree of similarity between the 4w and 8w timepoint. In contrast, in skin, the data show a high correlation between 1w and 4w, but lower correlation with the 8w timepoint. This is surprising (at least from the view of physical appearance of 1 week vs 4 week vs 8 week mice) and might indicate a sample mixup.
3. It is unclear which criteria were used to filter protein/peptide identifications. The authors claim to have quantified >11.000 proteins, but how many of those were e.g. identified only by a single peptide? Are those numbers referring to protein groups? Which protein interference strategy was used in DIA-NN?
4. Any validation of quantitative changes by other means (e.g. western blot, immunofluorescence etc.) is completely missing. The authors should at least validate some of their key findings with an independent experiment/technology, such as e.g. immunofluorescence on tissue slices.

Minor points:

1. Page1,Line19 : “from the same individuals” is misleading, as the organ harvesting can hardly be done on different times for the same individual.

2. Please provide analyses of CVs across biological replicates

3. Please provide reconstitution volume for samples before LC-MS analysis

4. Method descriptions are inconsistent, e.g. Homogenization and digestion reads like a copy-paste from a lab protocol.

5. The authors used “cystine carbamide methylation” as a fixed modification in their searches – please clarify, as this is not a standard term.

Reviewer #2 (Remarks to the Author):

The study by Wang et al uses state of the art mass spectrometry (MS) and data analysis tools to describe the proteomes of 10 different organs from male and female mice at 1, 4, and 8 weeks of age. Unbiased protein quantification at scale across entire organisms is clearly a new frontier and will generate resources for the community and advance our understanding of organism-wide physiology. The authors detect on average 6,000–8,000 proteins per organ and time point and they report proteins unique to organs, time points and sex or those common across organs. They highlight the importance of splicing during early postnatal development and in sexual dimorphism.

Overall, this is an impressive piece of work which is clearly analyzed and visualized with beautiful and simple illustrations. The authors do a nice job in highlighting individual pathways and proteins appropriate for an atlas-type study. There are some concerns and requests that should be addressed and which could help improve the study further.

1. One of my main concerns is the reproducibility of the findings in an independent cohort and with a different platform. While it would be unreasonable to see a complete replication of the entire dataset the authors should demonstrate that they can collect a couple of organs from an independent cohort, analyze it the same way and largely reproduce the findings. This independent validation is standard with other proteomic platforms, RNAseq, etc that generate atlas data. Again, a couple of organs in 5 mice should be sufficient.

The authors also need to show that some of the top proteins or pathways identified can be validated and quantified with an independent platform such as immunohistochemistry, Western blot, or ELISA/multiplex assays

2. Can the authors provide more information on how sample measurements are normalized against each other. A standard technique used in the field is to pool multiple barcoded samples in the same run to avoid batch effects (eg using TMT labeling). There is concern that differences in levels of proteins are in part the result of batch effects or run-to-run variation. In addition, can the authors show how much protein they extracted from each tissue. Is it possible that the overall concentration of proteins is simply higher with age but the relative levels of proteins (or some of them) do not change or go even in the opposite direction from what is reported?

The authors show levels of some “housekeeping” proteins which is reassuring and it would be nice to see similar graphs plotted for some of the top proteins across the various analyses.

3. The authors should avoid calling this an aging study or even a developmental study given that they have only three time points of which the 4 and 8 week timepoints seem to be quite similar. Maybe more emphasis could be placed on organ specific proteins instead. In retrospect, a postnatal developmental study would maybe analyze P1, P7, P14, P30, P60 and I’m not suggesting the authors should do this now. But, again, the data they have provide neither a deep biological insight into development nor into aging.

Reviewer #3 (Remarks to the Author):

The manuscript submitted by Wang et al. titled “Ten-organ developmental proteome atlas from infancy to adulthood mice” describes a comprehensive proteomics dataset investigating the changes in the proteome of mice across three time points (1-week, 4-weeks and 8-weeks), 10 organs and 2 genders in biological replicates, totaling 300 samples. The authors analyzed the proteomes of all organs from the same individual with 5 biological replicates from each gender at each time point. The authors further attempted to show some previously established biological insights using differential expression analysis among organs’ developmental stages as proof for the generated dataset. To the best of my knowledge, this is the most comprehensive study attempting to investigate developmental, organ and gender related changes on the proteome in mice and could be a very useful resource for future studies. As such, the project is a “tour-de-force”, without any follow up analysis, whereas the value largely is in the developed resource. However, a number of issues need to be resolved before publication may be possible.

Major Issues

- Since the most important output of this research is the generated dataset, one expects to see more even data exploration. From time to time, the analysis appears somewhat repetitive – DEPs and enrichment analysis in various dimensions. While I see that this is the core of the data, I would recommend elaborating on the interesting biological insights that the dataset shed light on or verified. Connected to this, some of the visualizations, particularly the heatmaps, can be visually improved. For example, I cannot tell whether e.g. the 115 age-related proteins found across all tissues show the same trend across age or not. It seems like some proteins go up in one tissue and down in another. Please extend the analysis by comparing proteins which show the same trend in all organs. Fuzzy clustering could help to further dissect potentially interesting proteins. Also, the authors show that proteins associated with the spliceosome appears to be related to age-related effects. Have the authors detected any relevant/interesting effects on protein sequences, e.g. loss or gain of peptides to pin point which proteins may be affected by this? Is there any evidence on protein level apart from an enriched spliceosome? Also, as far as I can tell, no supplementary tables are provided with e.g. the expression data across the 300 samples. Without access to processed, normalized and annotated data, in the form of at least supplementary tables, better even some online resources which would allow interaction with the data, the resource as such is not very useful and the lack of usable data may impair further usage substantially. I urge the authors to make the expression data with annotations available in addition to the raw data and search results.

- In a multi-batch experiment like this, removing batch effects is very important and influential on the findings. The authors acknowledge this step in the methods section (line 511) without providing any further details. Please elaborate. In addition, the methods seem to imply that each organ was measured as a “cohort”. Please elaborate on the measurement order, why this appears to be done in a per-organ basis and not (for statistical soundness) in a random order. How much does run-order influence the clustering analysis? Please check on the HeLa measurements, whether the clustering observed there is largely driven by measurement order and if this does negatively affect the value of the resource.

- The authors mention a DDA database and its analysis in the methods section (lines 480 and 495). However, this dataset is not mentioned anywhere. What does the dataset contain? What was its original purpose and why was it not used in the end?

- In e.g. lines 19 and 36, the sentence gives the wrong impression that samples for different time points were acquired from the same individual, which does not appear to be correct – see Methods. Please correct and tone down the wording in the abstract/introduction/discussion.

Minor Issues

- Please consult a language office to improve overall readability.
- The authors did not provide the coverage of mouse proteome or the average protein coverage. In addition, the authors could add some more basic statistics on
- The authors reported Pearson correlation coefficients (for example from line 115) without statistical significance test results, if they have done any. Considering the high dimensionality of the data, it may be beneficial to perform significance tests on the correlation measures.

- In the paragraph from line 110, the authors show the agreement of their data on single organs with already published single organ dataset in long, liver and stomach tissues (Supplementary Fig. 1c). However, as the authors mention in line 408, there are published data on the brain tissue. Why are the results not reported for this tissue?
- Since the data is generated in a DIA mode, the term “detected” is more suitable than “identified”. (e.g., line 21).
- There are some inconsistencies and mistakes in the figure legends. For example, in Supplementary Figure 1, the legends of subfigures c and d are swapped.
- There needs to be more conclusions and discussions for some of the results that the authors present in the manuscript. For example in line 148 (Fig. 2e), the authors mention that some of the differentially expressed proteins can form a protein interaction network which they present in the Fig. 2e. However, there is no discussion/interpretation about this network.
- In line 152, the authors mention that since only spliceosome pathways were enriched in their data, they must be essential in development. This sentence shows overconfidence in the produced data and disregards the importance of other proteins and pathways that are not detected in this study.
- There are some occurrences of the term “significant” through the manuscript (for example, line 205) for which the authors did not provide the results of the statistical tests, if they have done any. If no statistical tests have been performed, I would suggest using other terms like remarkable and to avoid the term significant.
- In lines 150 and 154, the same paragraph about the spliceosome is repeated with different references. Please avoid repetitions.
- The introduction mentions that DIA has no “bias to precursors in a pre-designed isolation ranges”. This is not true. Either remove or elaborate what was meant here.
- It seems that the term “adulthood” is defined differently for mouse in comparison to the manuscript. Please elaborate why and 8-week mouse is considered adult and not a more classic 3-6 months definition was used.
- Figure 1 highlights 1300 samples, whereas 300 were measured/generated. Please correct or elaborate.
- In Figure 1c, it is very difficult to differentiate the organs/time points. Please check alternative visualizations.
- Figure 1e was generated with or without clustering on the samples? Do samples generally cluster as expected? Please also add color legend for the time points.
- Supplementary Figure 6 is not annotated at all and thus the clustering shown does not “show” anything. Please add some form of additional annotation.
- Please add a colored label in Supplementary Figure 1 for gender. Also, why is the profile of e.g. heart, intestine and liver so different from the rest?
- Supplementary Figure 2 may point to the presence of sample mixups? E.g. Intestine 8-week female and muscle 4-week female. Please double check that all 300 samples are correctly labeled.

RESPONSES TO REVIEWER COMMENTS (NCOMMS-23-17406A)

Reviewers' comments are in *blue*, with authors' responses immediately following each Reviewer's comment and in black. Modifications to the manuscript are highlighted with a *yellow* background. Page numbers refer to the revised manuscript.

Response to Comments from Reviewer 1

Reviewer #1

In their manuscript entitled „Ten-organ developmental proteome atlas from infancy to adulthood mice“, Wang et al describe the results of a DIA-based proteomic study investigating the changes in proteomic composition in ten mouse tissues. Using a DIA approach the authors claim to have identified over 11.000 proteins and have identified candidate proteins that change with developmental stage. The presented dataset has high potential value for the community. While the rational of the study is well defined, and methods, data and evaluations are (with very minor exceptions) well documented in the supplementary datasets, the manuscript suffers from a number of potential major issues.

Major points:

1. Data quality: some of the muscle samples (all 5 biological replicates of the 8 week timepoint, male) show only a very weak correlation to any other samples of the entire study, this might indicate a problem in sample preparation – this in in contrast to the statement in line 108 and will impact the number of significantly regulated proteins between male and female mice.

We sincerely appreciate the reviewer's inquiry. To avoid problems caused by confusion in sample preparation or data analysis, we have scrutinized the original experimental records and found no potentially problematic records that could indicate sample mix-ups. Line 108 of our manuscript aspired to illustrate the correlation between the 5 biological replicates, and our data showed the Pearson correlation coefficient between 5 biological replicates of 8-week male muscle was: 0.972 (Supplementary Data 5).

As shown below, our data show a gradual decrease in inter-organ correlations with age (Supplementary Fig. 9). At 8 weeks, both muscle and brain showed weak correlations with other organs. This suggests to us that the weak correlation between muscle and other organs may be objective. However, further research and analysis are required to confirm this. Meanwhile, in line with our results, sexual dimorphism in muscle has been widely reported (*J Sport Health Sci. 2023 Jul;12(4):523-533.*; *J Sport Health Sci. 2023 Jul;12(4):523-533.*). In our study, we found a total of 769 differential protein groups between 8-week muscle males and females. Therefore, we believe that the weak correlation between muscle in males and females at 8-week is due to sex differences.

Supplementary Fig. 9 | Pearson correlation coefficient of ten organs proteomic data from infancy to adulthood. a, 1 week. b, 4-week. c, 8-week.

2. For most tissues (brain, heart, kidney, liver, lung) the correlation analyses indicate a high degree of similarity between the 4w and 8w timepoint. In contrast, in skin, the data show a high correlation between 1w and 4w, but lower correlation with the 8w timepoint. This is surprising (at least from the view of physical appearance of 1 week vs 4 week vs 8 week mice) and might indicate a sample mixup.

We greatly appreciate the comments from the reviewer. After reviewing our original experimental records, we could faithfully confirm no indications of potential sample mixing during either sample preparation or data analysis. Our proteomic data showed that the 4-week skin is more similar to the 1-week skin than to the 8-week skin, implying a shift in the mouse skin development at 4 weeks postnatally.

A literature published by Dekoninck S et al. used the murine tail and paw epidermis to unravel the mechanisms that mediate postnatal skin expansion (*Cell. 2020 Apr 30;181(3):604-620.e22.*). In their report, the tail surface grew linearly from P1 (postnatal day 1) to P30 and then slowly reached a plateau. On the other hand, the global growth kinetics of the tail slow down drastically after P30. Single-cell RNA Sequencing results shown the basal cells of the tail epidermis are more homogeneous during early postnatal development and that cellular heterogeneity begins around P30, at the time of the transition from a growing to a homeostatic mode of division, and further increased until mice reach their final size. The authors deduced that the clonal dynamics of interfollicular epidermis change around P30, which is consistent with our proteomics results.

3. It is unclear which criteria were used to filter protein/peptide identifications. The authors claim to have quantified >11,000 proteins, but how many of those were e.g. identified only by a single peptide? Are those numbers referring to protein groups? Which protein interference strategy was used in DIA-NN?

We truly appreciate the reviewer's inquiry and comments. We would like to elaborate our responses from the following 3 aspects:

1. Our protein groups (PGs) filtering strategy was that PGs with valid values in at least 4 out of 5 biological replicates per tissue per age per sex were retained, resulting in 11,533 PGs being obtained. For better readership clarity, we have also revised the manuscript.

Page 13, lines 550: Before further analysis, protein abundances (in log₂ scale) in each organ type didn't shown discrete batch effects in this study (Supplementary Fig. 2). Firstly, the filtration of reliable values, where proteins with at least four samples containing valid data out of the five biological replicates were retained. Coefficient of variance (CV) is used to assess the relative variability of the dataset⁶⁶ and median normalized abundance was calculated using data after log₂-transformation, followed by missing value imputation.

2. Based on the results obtained from the DIA_NN search data, we found a total of 706 protein groups containing a single peptide. And yes, 11,533 reported in this study is the number of protein groups. The DIA_NN output precursors data and PGs data have been deposited to the ProteomeXchange and reviewer can now freely access the data through the Username: reviewer_pxd041400@ebi.ac.uk; Password: RQMHZ2YZ.

Based on the reviewer's comment, we have also modified the "proteins" in the original manuscript to "protein groups (PGs)" to be more precise.

Page 1, lines 20: We detected and quantified 11,533 protein groups (PGs) across the 10 organs and obtained 115 age-related differentially expressed PGs that were co-expressed in all organs from infancy to adulthood.

3. The search software DIA_NN protein interference strategy is described in details as below in a previous literature:

"The DIA-NN workflow starts with a peptide-centric approach based on a collection of precursor ions which can be automatically generated by DIA-NN in silico from a protein sequence database. DIA-NN then generates a library of negative controls (that is, decoy precursors), extracts a chromatogram for each target or decoy precursor and identifies putative elution peaks comprised of the precursor and fragment ion elution profiles in the vicinity of the putative retention time of the precursor. Each elution peak is then described by a set of scores that reflect peak characteristics, including co-elution of fragment ions, mass accuracy or similarity between observed and reference (library) spectra. In total, DIA-NN calculates 73 peak scores in the various steps of the workflow. The best candidate peak is then selected per precursor using iterative training of a linear classifier, which allows the calculation of a single discriminant score for each peak." (*Nat Methods*. 2020 Jan;17(1):41-44.). On the other hand, in order to avoid false identifications and unreliable quantification "DIA-NN evaluates the degree of

interference between multiple precursors initially matched to the same retention time, and, if the interference is deemed significant enough, reports only those best supported by the data as identified, improving the identification performance at strict false discovery rate (FDR) thresholds.” (*Nat Methods*. 2020 Jan;17(1):41-44.)

4. Any validation of quantitative changes by other means (e.g. western blot, immunofluorescence etc.) is completely missing. The authors should at least validate some of their key findings with an independent experiment/technology, such as e.g. immunofluorescence on tissue slices.

We appreciate the reviewer for the insightful comments. Per the reviewer’s request, we have performed Western blot experiments to verify the spliceosome proteins (Q9CWZ3 gene name Rbm8a and P26369 gene name U2af2) in ten organs with age. These new results are consistent with the mass spectrometry findings. These results were added to the manuscript as supplementary Fig. 5.

Page 5, lines 174: Q9CWZ3 (gene name Rbm8a) plays a critical role in the development of inhibitory neurons and Rbm8a perturbation leads to profound cortical deficits³⁵. Protein P26369, also known as U2af2, can regulate Wilms' tumor gene 1 (WT1) to jointly play an important role in pre-mRNA splicing³⁶. WT1 is crucial for normal kidney development, and mutations can cause kidney tumors and lead to Denys Drash or Frasier syndromes³⁷. In our study, expression of RBM8A in the brain and U2AF2 in the kidney decreased significantly from infancy to adulthood (Fig. 2j). Western blot experiments confirmed the trend of decreasing expression of the splice variants with age, which is consistent with the mass spectrometry experiment (Supplementary Fig. 5). All original images can be viewed from ProteomeXchange.

35 McSweeney, C. et al. Full function of exon junction complex factor, Rbm8a, is critical for interneuron development. *Transl Psychiatry* 10, 379 (2020).

36 Davies, R. C. et al. WT1 interacts with the splicing factor U2AF65 in an isoform-dependent manner and can be incorporated into spliceosomes. *Genes Dev* 12, 3217-3225 (1998).

37 Hammes, A. et al. Two splice variants of the Wilms' tumor 1 gene have distinct functions during sex determination and nephron formation. *Cell* 106, 319-329 (2001).

Supplementary Fig. 5 | Western blots results of spliceosome protein: RBM8A(a), and U2AF2(b). Ratio = target protein gray value: GAPDH gray value. (mean \pm SD; n=3). In the box plot, the median is represented by the center line and the box boundaries represent the first and third quartiles. * p -value < 0.05, ** p -value < 0.01, *** p -value < 0.001 according to a one-way ANOVA. n=3 biologically independent mice.

Further, we have also performed additional experiments on the timsTOF flex (Bruker) mass spectrometry platform to validate our results. Our new results show that the proteomic data from two mass spectrometers with different detection principles are in high agreement. These new results were added to the manuscript as supplementary Fig. 10.

Page 7, lines 289: The instrument we used in this study was an EASY nLC 1200 coupled with QE HF-X system (Thermo Fisher Scientific). We further validated our brain-unique DEPs results on an additional timsTOF flex (Bruker) mass spectrometry platform. The results showed that both mass spectrometry detected 6,671 PGs in the

brain, about 78% of the total amount (Supplementary Fig. 10a, Supplementary Data 15). Heatmaps and expression trends of the two mass spectrometry platforms regarding brain-unique age-related DEPs (ANOVA adjust p-value < 0.05) were shown in Supplementary Fig. 10b. The number of age-related down- and up-regulated DEPs detected by both platforms was 221 and 343, respectively (Supplementary Fig. 10c). PCA analysis also showed the consistency of the mass spectrometry data generated by the two different platforms (Supplementary Fig. 10d). Moreover, we generated quantile–quantile (Q–Q) plots⁵² to compare the P values reported of the data detected by the two mass spectrometers with the normalized rank P values. that represent the uniform distribution to illustrate the consistency of the data obtained by the two instruments (Supplementary Fig. 10e). The data points exhibit an evenly distributed pattern along the diagonal, indicating strong reproducibility of the detected data across both platforms.

52 Galwey, N. W. A Q–Q plot aids interpretation of the false discovery rate. *Biom J* 65, e2100309 (2023).

Supplementary Fig. 10 | Brain-unique age-related DEPs validation. **a**, Venn diagram shows brain tissue proteomic data obtained from two mass spectrometry instruments, Orbitrap QE HF-X system (Thermo Fisher Scientific) and timsTOF (Bruker). **b**, Heatmap of brain-unique age-related DEPs by semi-supervised hierarchical clustering. Top: data from timsTOF (Bruker); Bottom: data from Orbitrap (Thermo Fisher Scientific). The protein expression trends fall into the down-regulated module and the up-regulated module. **c**, Venn diagram shows down-regulated DEPs data (top) and up-regulated protein data (bottom) obtained from two mass spectrometers. **d**, PCA analyses of proteomic data from Orbitrap QE and timsTOF LC-MS platform. **e**, Q–Q plots to compare the P values reported of the data detected by the two mass spectrometers with the normalized rank P values that represent the uniform distribution. The data points exhibit an evenly distributed pattern along the diagonal, indicating strong reproducibility of the detected data across both

platforms.

Minor points:

1. Page1, Line19 : “from the same individuals” is misleading, as the organ harvesting can hardly be done on different times for the same individual.

Thanks for the comments and we do apologize for the misunderstandings in the original manuscript. We have revised the phrase “self-multi-organs” to “multi-organ” throughout the manuscript to be more precise:

Page 1, lines 17: Herein, we present a comprehensive proteomic analysis of ten mouse organs (brain, heart, lung, liver, kidney, spleen, stomach, intestine, muscle and skin) at three crucial developmental stages (1-week, 4-week and 8-week after birth) by data-independent acquisition mass spectrometry.

Page 1, lines 25: This multi-organ proteome atlas provides a fundamental baseline for understanding the molecular mechanisms underlying organ development and maturation in early-life.

Page 2, lines 39: To date, the molecular profile of early-life multi-organ comparisons still remains scarce.

Page 2, lines 52: Mapping the multi-organ development proteome atlas can therefore detect the molecular regulators and effectors of developmental biology physiological activities.

Page 10, lines 400: This organism-wide characterization of early-life dynamic proteomics atlas fills in the knowledge gap for early-life multi-organ development and maturation.

Page 10, lines 402: Systematic and simultaneous detection of proteomic data from multiple organs under the same conditions could provide a more comprehensive picture of in vivo early life development.

2. Please provide analyses of CVs across biological replicates

We appreciate the inquiry. The median CV of the protein quantification in the biological replicate samples is shown below and added to the manuscript as supplementary Fig. 4. Protein quantification in the biological replicate samples showed median CVs below 0.3, except for 8-week female intestine and 4-week female muscle. The correlated contents were also updated in the manuscript:

Page 3, lines 104: We found very high correlation between the biologically replicates samples of organs of the same sex averaged 0.974, again indicating stability of the data (Fig. 1d Supplementary Data 3). The coefficient of variation (CV) is used to assess the relative variability of the dataset, with higher CV values representing more dispersion³³. In our study, protein quantification in the most biological replicate samples showed median coefficient of variance (CV) below 0.3 (Supplementary Fig. 3, Supplementary Data 4). Due to individual variation, the intestinal and muscle samples exhibited slightly higher coefficients of variation (CVs).

33 Botta-Dukát, Z. Quartile coefficient of variation is more robust than CV for traits calculated as a ratio. *Sci Rep* 13, 4671 (2023).

Supplementary Fig. 3 | Coefficient of variation (CV) across biological replicates.

The median CV of the proteomics data was calculated by the normalized of quantified proteins. **a**, brain. **b**, heart. **c**, intestine. **d**, kidney. **e**, liver. **f**, lung. **g**, muscle. **h**, skin. **i**, spleen. **j**, stomach. In the box plot, the median is represented by the center line and the box boundaries represent the first and third quartiles.

Page 9, lines 449: To gain further insight into the sex influence during organ development and maturation, the Student's t-test was performed to analyze the sex-related DEPs changes between males and females in 1-week infancy mice (Fig. 6a), 4-week young mice (Fig. 6b) and 8-week adulthood mice (Fig. 6c, Supplement Data 17).

3. Please provide reconstitution volume for samples before LC-MS analysis

Thanks for the inquiry. Before LC-MS analysis, each sample was taken out from the -80°C freezer and resuspended separately using 60 µl of 0.1% FA water, followed by shaking and sonication to fully dissolve peptide fragments. After quantification of peptide concentration using Nanodrop, the sample concentration was adjusted to 500 ng/µl by adding 0.1% FA water. After transferring to the injection vial, each sample was injected with 1 µl (equivalent to 500 ng) for mass spectrometry analysis. These related elaborations are also updated in the manuscript:

Page 12, lines 494: MS analysis was performed on an EASY nLC1200 coupled with QE HF-X system (Thermo Fisher Scientific). The samples were resuspended in 0.1% FA, and adjust the final concentration to 0.5 µg/µl. After spiked with iRT (Biognosys), 0.5 µg of each sample was loaded onto 2 cm × 75 µm homemade reversed-phase C18 column (particle size, 3 µm; pore size, 100Å. Thermo Fisher Scientific) and separated by 25 cm × 75 µm analytical column (particle size, 2 µm; pore size, 100 Å. Thermo Fisher Scientific).

4. Method descriptions are inconsistent, e.g. Homogenization and digestion reads like a copy-paste from a lab protocol.

We truly appreciate the reviewer's comments, and we have further revised the methods section to elaborate our approach as clearly as possible.

Page 12, lines 472: Homogenization and digestion. There are in total 300 samples from 10 organs from 30 individual mice. 5 females and 5 males at each time point. The sample preparation method was as described Filter-aided sample preparation (FASP)³¹. The detailed descriptions were as follows:

About 10~20mg of tissue were weighed and rapidly frozen in liquid nitrogen, then ground in a mill at 60 Hz and -20°C for 40 seconds. Next, lysate containing protease inhibitor was added to tissue at a concentration of 20 µl/mg before grinding again. After sonication on ice, the remaining debris was removed by centrifugation at 10,000 g for 20 min and the supernatant lysates were quantified by BCA assay (Thermo Fisher Scientific).

For digestion, a total of 100 µg protein per sample was precipitated in acetone overnight

at -20°C. The protein pellet was obtained after centrifugation at 14000 g for 20 min, and the pellet was washed twice with acetone. After discarding the supernatant and evaporating the residual acetone, the pellet was resuspended in 7M guanidine hydrochloride. The protein solution was reduced with 10mM dithiothreitol (DTT) at 55°C for 1 hour and alkylated by 20 mM iodoacetamide (IAA) for 30min in the dark. Samples were washed with ammonium bicarbonate on 10 kDa ultrafiltration concentrators (Sartorius) to purify the protein. Finally, the trypsin (Promega) digestion was performed overnight at 37°C using a 50:1 protein: protease ratio (w/w). After termination of digestion by the addition of a final concentration of 1% formic acid (FA), the peptides were desalted using a C18 spin column (Monospin) according to the manufacturer's instructions. Peptides eluted were dried under vacuum and stored at -80°C until subsequent use for liquid chromatography tandem mass spectrometry (LC-MS/MS) analysis.

31 Wiśniewski, J. R., Zougman, A., Nagaraj, N. & Mann, M. Universal sample preparation method for proteome analysis. *Nat Methods* 6, 359-362 (2009).

5. The authors used “cystine carbamide methylation” as a fixed modification in their searches – please clarify, as this is not a standard term.

We appreciate the reviewer's detailed comments and apologize for any confusion due to our typo. We have made corrections to the manuscript:

Page 13, lines 534: Specifically, N-terminal acetylation and methionine oxidation were set as variable modifications, whereas cysteine carbamidomethylation was set as fixed modification.

Reviewer #2

The study by Wang et al uses state of the art mass spectrometry (MS) and data analysis tools to describe the proteomes of 10 different organs from male and female mice at 1, 4, and 8 weeks of age. Unbiased protein quantification at scale across entire organisms is clearly a new frontier and will generate resources for the community and advance our understanding of organism-wide physiology. The authors detect on average 6,000–8,000 proteins per organ and time point and they report proteins unique to organs, time points and sex or those common across organs. They highlight the importance of splicing during early postnatal development and in sexual dimorphism.

Overall, this is an impressive piece of work which is clearly analyzed and visualized with beautiful and simple illustrations. The authors do a nice job in highlighting individual pathways and proteins appropriate for an atlas-type study. There are some concerns and requests that should be addressed and which could help improve the study further.

1. One of my main concerns is the reproducibility of the findings in an independent cohort and with a different platform. While it would be unreasonable to see a complete replication of the entire dataset the authors should demonstrate that they can collect a couple of organs from an independent cohort, analyze it the same way and largely reproduce the findings. This independent validation is standard with other proteomic platforms, RNAseq, etc that generate atlas data. Again, a couple of organs in 5 mice should be sufficient.

The authors also need to show that some of the top proteins or pathways identified can be validated and quantified with an independent platform such as immunohistochemistry, Western blot, or ELISA/multiplex assays

We truly appreciate the reviewer for the insightful comments. To better address the reviewer's inquiry, we have performed additional experiments on timsTOF flex (Bruker) mass spectrometry platform to validate our findings. The results show that the proteomic data from two mass spectrometers with different detection principles are in high agreement. These new results added to the manuscript as supplementary Fig. 10.

Page 7, lines 289: The instrument we used in this study was an EASY nLC 1200 coupled with QE HF-X system (Thermo Fisher Scientific). We further validated our brain-unique DEPs results on an additional timsTOF flex (Bruker) mass spectrometry platform. The results showed that both mass spectrometry detected 6,671 PGs in the brain, about 78% of the total amount (Supplementary Fig. 10a, Supplementary Data 15). Heatmaps and expression trends of the two mass spectrometry platforms regarding brain-unique age-related DEPs (ANOVA adjust p-value < 0.05) were shown in Supplementary Fig. 10b. The number of age-related down- and up-regulated DEPs detected by both platforms was 221 and 343, respectively (Supplementary Fig. 10c). PCA analysis also showed the consistency of the mass spectrometry data generated by the two different platforms (Supplementary Fig. 10d). Moreover, we generated quantile–quantile (Q–Q) plots⁵² to compare the P values reported of the data detected by the two mass spectrometers with the normalized rank P values, that represent the uniform distribution to illustrate the consistency of the data obtained by the two instruments (Supplementary Fig. 10e). The data points exhibit an evenly distributed pattern along the diagonal, indicating strong reproducibility of the detected data across both platforms.

52 Galwey, N. W. A Q–Q plot aids interpretation of the false discovery rate. *Biom J* 65, e2100309 (2023).

Supplementary Fig. 10 | Brain-unique age-related DEPs validation. **a**, Venn diagram shows brain tissue proteomic data obtained from two mass spectrometry instruments, Orbitrap QE HF-X system (Thermo Fisher Scientific) and timsTOF (Bruker). **b**, Heatmap of brain-unique age-related DEPs by semi-supervised hierarchical clustering. Top: data from timsTOF (Bruker); Bottom: data from Orbitrap (Thermo Fisher Scientific). The protein expression trends fall into the down-regulated module and the up-regulated module. **c**, Venn diagram shows down-regulated DEPs data (top) and up-regulated protein data (bottom) obtained from two mass spectrometers. **d**, PCA analyses of proteomic data from Orbitrap QE and timsTOF LC-MS platform. **e**, Q-Q plots to compare the P values reported of the data detected by the two mass spectrometers with the normalized rank P values that represent the uniform distribution. The data points exhibit an evenly distributed pattern along the diagonal, indicating strong reproducibility of the detected data across both platforms.

Meanwhile, per the reviewer's request, we have also performed Western blot experiments to verify the spliceosome proteins (Q9CWZ3 gene name Rbm8a and P26369 gene name U2af2) in organs with age. These new results are consistent with the mass spectrometry experiments. The WB results added to the manuscript as supplementary Fig. 5.

Page 5, lines 174: Q9CWZ3 (gene name Rbm8a) plays a critical role in the development of inhibitory neurons and Rbm8a perturbation leads to profound cortical deficits³⁵. Protein P26369, also known as U2af2, can regulate Wilms' tumor gene 1 (WT1) to jointly play an important role in pre-mRNA splicing³⁶. WT1 is crucial for normal kidney development, and mutations can cause kidney tumors and lead to Denys Drash or Frasier syndromes³⁷. In our study, expression of RBM8A in the brain and U2AF2 in the kidney decreased significantly from infancy to adulthood (Fig. 2j). Western blot experiments confirmed the trend of decreasing expression of the splice variants with age, which is consistent with the mass spectrometry experiment (Supplementary Fig. 5). All original images can be viewed from ProteomeXchange.

35 McSweeney, C. et al. Full function of exon junction complex factor, Rbm8a, is critical for interneuron development. *Transl Psychiatry* 10, 379 (2020).

36 Davies, R. C. et al. WT1 interacts with the splicing factor U2AF65 in an isoform-dependent manner and can be incorporated into spliceosomes. *Genes Dev* 12, 3217-3225 (1998).

37 Hammes, A. et al. Two splice variants of the Wilms' tumor 1 gene have distinct

functions during sex determination and nephron formation. Cell 106, 319-329 (2001).

Supplementary Fig. 5 | Western blots results of spliceosome protein: RBM8A(a), and U2AF2(b). Ratio = target protein gray value: GAPDH gray value. (mean \pm SD; n=3). In the box plot, the median is represented by the center line and the box boundaries represent the first and third quartiles. * p -value < 0.05, ** p -value < 0.01, *** p -value < 0.001 according to a one-way ANOVA. n=3 biologically independent mice.

2. Can the authors provide more information on how sample measurements are normalized against each other. A standard technique used in the field is to pool multiple barcoded samples in the same run to avoid batch effects (eg using TMT labeling). There is concern that differences in levels of proteins are in part the result of batch effects or run-to-run variation. In addition, can the authors show how much protein they extracted from each tissue. Is it possible that the overall concentration of proteins is simply higher

with age but the relative levels of proteins (or some of them) do not change or go even in the opposite direction from what is reported?

The authors show levels of some “housekeeping” proteins which is reassuring and it would be nice to see similar graphs plotted for some of the top proteins across the various analyses.

Thanks for the detailed questions. And we would like to break down our responses in the following two parts:

1. We sincerely appreciate the detailed suggestion and inquiry about the batch effect from the reviewer. We would like to provide the following four aspects to confirm that the results of this study are not influenced by batch effects and that the conclusions are indeed reliable.

Firstly, for experimental manipulation, to ensure the stability of mass spectrometry analysis, we spiked iRT (indexed retention time), which is a mixture of 11 synthetic peptides that do not exist naturally, in the samples for analysis. The inclusion of iRT peptides enables the generation of high-quality spectral libraries and accurate normalization in DIA or SWATH detection, thereby enhancing the reliability of proteomic data and reduced the run-to-run variation (*Proteomics*. 2012 Apr;12(8):1111-21.).

Secondly, based on Čuklina et al.'s description of batch effects (*Mol Syst Biol*. 2021 Aug;17(8): e10240.), intensity boxplots are extremely powerful at the initial assessment stage. Simultaneously, with reference to the judgment methods proposed by Jiang et al. (*Cell*. 2020 Oct 1;183(1):269-283.e19.), we utilized log₂-transformed protein abundance data to determine the presence of batch effects in our cohort before conducting data analysis. No obvious discrete batch effects were observed in the intensity plot of protein abundances in each age group for each organ. The log₂-transformed protein abundance data were added to the manuscript as supplementary Fig. 2, and we have also revised the description in the manuscript:

Page 3, lines 104: Our protein abundance data, which were transformed using a logarithm base 2, demonstrate stability across different batches (Supplementary Fig. 2).

Page 13, lines 550: Before further analysis, protein abundances (in log₂ scale) in each organ type didn't show discrete batch effects in this study (Supplementary Fig. 2).

Supplementary Fig. 2 | Protein abundances (in log₂ scale) in each age group for each organ. In the box plot, the median is represented by the center line and the box boundaries represent the first and third quartiles.

Thirdly, we additionally added a proteomics cohort containing 40 samples to validate whether the measurement order has a batch effect. These results showed that the measurement order doesn't affect the final sample clustering resulting (Supplementary Fig. 1e).

Supplementary Fig. 1e | Proteomic atlas of ten organs from infancy to adulthood mice. e, After order and disorder detection of a total of 40 samples from the brain, kidney, liver and spleen, the mass spectrometry results were analyzed by PCA.

The related contents are also provided in the revised manuscript as below:

Page 3, lines 120: The multi-organ study also revealed clustering between different organs. To avoid run-order influence in the clustering analysis, we assessed whether the order of mass spectrometry detection would lead to batch effects. A total of 40 samples from brain, kidney, liver and spleen were performed a random mass spectrometry. We found that regardless of the order or disorder of LC-MS measurement, the samples tight clustered according to tissue types, indicating that in our study the detection order does not affect the mass spectrometry data and cause batch effects (Supplementary Fig. 1e, Supplementary Data 5).

Finally, WB validation experiments (Supplementary Fig. 5) and mass spectrometry verification (Supplementary Fig. 10) collectively confirmed our findings and illustrated the reliability of our analysis.

Supplementary Fig. 5 | Western blots results of spliceosome protein: RBM8A(a), and U2AF2(b). Ratio = target protein gray value: GAPDH gray value. (mean \pm SD; n=3). In the box plot, the median is represented by the center line and the box boundaries represent the first and third quartiles. * p -value < 0.05, ** p -value < 0.01, *** p -value < 0.001 according to a one-way ANOVA. n=3 biologically independent mice.

Supplementary Fig. 10 | Brain-unique age-related DEPs validation. **a**, Venn diagram shows brain tissue proteomic data obtained from two mass spectrometry instruments, Orbitrap QE HF-X system (Thermo Fisher Scientific) and timsTOF (Bruker). **b**, Heatmap of brain-unique age-related DEPs by semi-supervised hierarchical clustering. Top: data from timsTOF (Bruker); Bottom: data from Orbitrap (Thermo Fisher Scientific). The protein expression trends fall into the down-regulated module and the up-regulated module. **c**, Venn diagram shows down-regulated DEPs data (top) and up-regulated protein data (bottom) obtained from two mass spectrometers. **d**, PCA analyses of proteomic data from Orbitrap QE and timsTOF LC-MS platform. **e**, Q-Q plots to compare the P values reported of the data detected by the two mass spectrometers with the normalized rank P values that represent the uniform distribution. The data points exhibit an evenly distributed pattern along the diagonal, indicating strong reproducibility of the detected data across both platforms.

2. Thanks to the reviewer for inquiring about the extract from each tissue. Our protein extraction strategy was to weigh 10-20 mg of tissue, add lysis buffer (RIPA) at a ratio of 1 mg : 20 μ l, and then milled to extract the proteins. The BCA assay data for each sample was uploaded in the OneDriver (<https://1drv.ms/x/s!AsvSofUEIDd3hzG7kx0KIK7C1CRt?e=9Dc0Mv>). All analyzed data will be uploaded to both ProteomeXchange and figshare data sharing platforms for readers' access at the same time. ProteomeXchange reviewer account information is also available (Username: reviewer_pxd041400@ebi.ac.uk; Password: RQMZH2YZ).

If reviewers do not find the data, please feel free to let us know through the editors or any other means so that we can provide the data as soon as possible.

After tissue lysis, 100 ug of protein is digested into peptides, and finally 500 ng of peptides go to mass spectrometry detection. This process is equivalent to randomly sampling from the total protein content within the tissues, and the final result is not influenced by the absolute total protein amount in the tissues.

3. The authors should avoid calling this an aging study or even a developmental study given that they have only three time points of which the 4 and 8 week timepoints seem to be quite similar. Maybe more emphasis could be placed on organ specific proteins instead. In retrospect, a postnatal developmental study would maybe analyze P1, P7, P14, P30, P60 and I'm not suggesting the authors should do this now. But, again, the data they have provide neither a deep biological insight into development nor into aging.

The authors really appreciate the reviewer's suggestions. Based on the reviewers' comments, we have revised the title of this study to "Ten-organ proteome atlas from infancy to adulthood mice". Through mass spectrometry experiments on different platforms, we emphasize the results of brain-unique age-related DEPs. Moreover, the discussion on organ-unique age-related DEPs has also been expanded.

Page 7, lines 289: The instrument we used in this study was an EASY nLC 1200 coupled with QE HF-X system (Thermo Fisher Scientific). We further validated our brain-unique DEPs results on an additional timsTOF flex (Bruker) mass spectrometry platform. The results showed that both mass spectrometry detected 6,671 PGs in the brain, about 78% of the total amount (Supplementary Fig. 10a, Supplementary Data 15). Heatmaps and expression trends of the two mass spectrometry platforms regarding brain-unique age-related DEPs (ANOVA adjust p-value < 0.05) were shown in Supplementary Fig. 10b. The number of age-related down- and up-regulated DEPs detected by both platforms was 221 and 343, respectively (Supplementary Fig. 10c). PCA analysis also showed the consistency of the mass spectrometry data generated by the two different platforms (Supplementary Fig. 10d). Moreover, we generated quantile-quantile (Q-Q) plots⁵² to compare the P values reported of the data detected by the two mass spectrometers with the normalized rank P values. that represent the uniform distribution to illustrate the consistency of the data obtained by the two instruments (Supplementary Fig. 10e). The data points exhibit an evenly distributed

pattern along the diagonal, indicating strong reproducibility of the detected data across both platforms.

52 Galwey, N. W. A Q-Q plot aids interpretation of the false discovery rate. *Biom J* 65, e2100309 (2023).

Supplementary Fig. 10 | Brain-unique age-related DEPs validation. **a**, Venn diagram shows brain tissue proteomic data obtained from two mass spectrometry instruments, Orbitrap QE HF-X system (Thermo Fisher Scientific) and timsTOF (Bruker). **b**, Heatmap of brain-unique age-related DEPs by semi-supervised hierarchical clustering. Top: data from timsTOF (Bruker); Bottom: data from Orbitrap (Thermo Fisher Scientific). The protein expression trends fall into the down-regulated module and the up-regulated module. **c**, Venn diagram shows down-regulated DEPs data (top) and up-regulated protein data (bottom) obtained from two mass spectrometers. **d**, PCA analyses of proteomic data from Orbitrap QE and timsTOF LC-MS platform. **e**, Q-Q plots to compare the P values reported of the data detected by the two mass spectrometers with the normalized rank P values that represent the uniform distribution. The data points exhibit an evenly distributed pattern along the diagonal, indicating strong reproducibility of the detected data across both platforms.

Page 11, lines 434: This study has also proposed organ-unique age-related DEPs, which can contribute to a better parsing the independent development of single-organ and the overall development of organisms. Further associations, these findings can establish a bridge between individual organs and the organic whole, and are relevant for studies of organoids, organ-specific diseases, and systemic diseases. Interestingly, the inter-age correlation of the skin is different from other organs, and our proteomics

data show that 4-week-old skin is closer to 1-week-old skin. This result is similar to that of Dekoninck et al. in murine tail and paw epidermis³⁴, both of which indicate that mouse skin undergoes rapid development until 4 weeks, after which it enters a period of maturation and stabilization. All of these phenomena pointed to commonalities in organ development as well as many organ-unique characteristics.

34 Dekoninck, S. *et al.* Defining the Design Principles of Skin Epidermis Postnatal Growth. *Cell* 181, 604–620. e622 (2020).

Reviewer #3

The manuscript submitted by Wang et al. titled “Ten-organ developmental proteome atlas from infancy to adulthood mice” describes a comprehensive proteomics dataset investigating the changes in the proteome of mice across three time points (1-week, 4-weeks and 8-weeks), 10 organs and 2 genders in biological replicates, totaling 300 samples. The authors analyzed the proteomes of all organs from the same individual with 5 biological replicates from each gender at each time point. The authors further attempted to show some previously established biological insights using differential expression analysis among organs’ developmental stages as proof for the generated dataset. To the best of my knowledge, this is the most comprehensive study attempting to investigate developmental, organ and gender related changes on the proteome in mice and could be a very useful resource for future studies. As such, the project is a “tour-de-force”, without any follow up analysis, whereas the value largely is in the developed resource. However, a number of issues need to be resolved before publication may be possible.

Major Issues

- Since the most important output of this research is the generated dataset, one expects to see more even data exploration. From time to time, the analysis appears somewhat repetitive – DEPs and enrichment analysis in various dimensions. While I see that this is the core of the data, I would recommend elaborating on the interesting biological insights that the dataset shed light on or verified. Connected to this, some of the visualizations, particularly the heatmaps, can be visually improved. For example, I cannot tell whether e.g. the 115 age-related proteins found across all tissues show the same trend across age or not. It seems like some proteins go up in one tissue and down in another. Please extend the analysis by comparing proteins which show the same trend in all organs. Fuzzy clustering could help to further dissect potentially interesting proteins. Also, the authors show that proteins associated with the spliceosome appears to be related to age-related effects. Have the authors detected any relevant/interesting effects on protein sequences, e.g. loss or gain of peptides to pin point which proteins

may be affected by this? Is there any evidence on protein level apart from an enriched spliceosome? Also, as far as I can tell, no supplementary tables are provided with e.g. the expression data across the 300 samples. Without access to processed, normalized and annotated data, in the form of at least supplementary tables, better even some online resources which would allow interaction with the data, the resource as such is not very useful and the lack of usable data may impair further usage substantially. I urge the authors to make the expression data with annotations available in addition to the raw data and search results.

We truly appreciate the reviewer's expertise suggestions and comments. We would like to break down our responses in the following four aspects:

1. To better address the reviewer's inquiry and to elaborate on the interesting biological insights, we have performed additional experiments on another LC-MS platform, namely timsTOF flex (Bruker) mass spectrometry platform, to validate our results. Results added to the manuscript as supplementary Fig. 10. These new results show that the proteomic data from two mass spectrometers with different detection principles are in high agreement. The brain-unique age-related DEPs proposed in this study are emphasized. The related contents are also updated in the revised manuscript as below:

Page 7, lines 289: The instrument we used in this study was an EASY nLC 1200 coupled with QE HF-X system (Thermo Fisher Scientific). We further validated our brain-unique DEPs results on an additional timsTOF flex (Bruker) mass spectrometry platform. The results showed that both mass spectrometry detected 6,671 PGs in the brain, about 78% of the total amount (Supplementary Fig. 10a, Supplementary Data 15). Heatmaps and expression trends of the two mass spectrometry platforms regarding brain-unique age-related DEPs (ANOVA adjust p-value < 0.05) were shown in Supplementary Fig. 10b. The number of age-related down- and up-regulated DEPs detected by both platforms was 221 and 343, respectively (Supplementary Fig. 10c). PCA analysis also showed the consistency of the mass spectrometry data generated by the two different platforms (Supplementary Fig. 10d). Moreover, we generated quantile–quantile (Q–Q) plots⁵² to compare the P values reported of the data detected by the two mass spectrometers with the normalized rank P values. that represent the uniform distribution to illustrate the consistency of the data obtained by the two instruments (Supplementary Fig. 10e). The data points exhibit an evenly distributed pattern along the diagonal, indicating strong reproducibility of the detected data across both platforms.

Supplementary Fig. 10 | Brain-unique age-related DEPs validation. **a**, Venn diagram shows brain tissue proteomic data obtained from two mass spectrometry instruments, Orbitrap QE HF-X system (Thermo Fisher Scientific) and timsTOF (Bruker). **b**, Heatmap of brain-unique age-related DEPs by semi-supervised hierarchical clustering. Top: data from timsTOF (Bruker); Bottom: data from Orbitrap (Thermo Fisher Scientific). The protein expression trends fall into the down-regulated module and the up-regulated module. **c**, Venn diagram shows down-regulated DEPs data (top) and up-regulated protein data (bottom) obtained from two mass spectrometers. **d**, PCA analyses of proteomic data from Orbitrap QE and timsTOF LC-MS platform. **e**, Q-Q plots to compare the P values reported of the data detected by the two mass spectrometers with the normalized rank P values that represent the uniform distribution. The data points exhibit an evenly distributed pattern along the diagonal, indicating strong reproducibility of the detected data across both platforms.

Furthermore, we have also performed Western blot experiments to verify our findings. The new WB results were added to the manuscript as supplementary Fig. 5.

Page 5, lines 174: Q9CWZ3 (gene name Rbm8a) plays a critical role in the development of inhibitory neurons and Rbm8a perturbation leads to profound cortical deficits³⁵. Protein P26369, also known as U2af2, can regulate Wilms' tumor gene 1 (WT1) to jointly play an important role in pre-mRNA splicing³⁶. WT1 is crucial for

normal kidney development, and mutations can cause kidney tumors and lead to Denys Drash or Frasier syndromes³⁷. In our study, expression of RBM8A in the brain and U2AF2 in the kidney decreased significantly from infancy to adulthood (Fig. 2j). Western blot experiments confirmed the trend of decreasing expression of the splice variants with age, which is consistent with the mass spectrometry experiment (Supplementary Fig. 5). All original images can be viewed from ProteomeXchange.

35 McSweeney, C. et al. Full function of exon junction complex factor, Rbm8a, is critical for interneuron development. *Transl Psychiatry* 10, 379 (2020).

36 Davies, R. C. et al. WT1 interacts with the splicing factor U2AF65 in an isoform-dependent manner and can be incorporated into spliceosomes. *Genes Dev* 12, 3217-3225 (1998).

37 Hammes, A. et al. Two splice variants of the Wilms' tumor 1 gene have distinct functions during sex determination and nephron formation. *Cell* 106, 319-329 (2001).

Supplementary Fig. 5 | Western blots results of spliceosome protein: RBM8A(a), and U2AF2(b). Ratio = target protein gray value: GAPDH gray value. (mean \pm SD; n=3). In the box plot, the median is represented by the center line and the box

boundaries represent the first and third quartiles. * p -value < 0.05, ** p -value < 0.01, *** p -value < 0.001 according to a one-way ANOVA. $n=3$ biologically independent mice.

2. Following the reviewers' suggestions, we have adjusted the heatmap presentation of Figure 2 and added the results of Mfuzz clustering to better show the trend of protein expression with age.

Fig. 2 | Age-related differential proteins across 10 organs from infancy to adulthood. **a**, The outer circle number indicates age-related differentially expressed proteins (DEPs) in each organ (the ANOVA cutoff of DEPs was set at adjusted p -value < 0.05). 115 age-related DEPs were detected by cross-comparing DEPs between 10 organs. **b**, Enrichment of KEGG pathway subsets of age-related DEPs in different organs. The size of the circle represents the $-\log_{10}(p\text{-value})$ and the color represents the fold enrichment. **c**, t-distributed stochastic neighbor embedding (t-SNE) visualization of the 115 DEPs to cluster organs and ages. **d**, Heatmap of the 115 DEPs expressed across different organs and ages. The heatmap is based on the abundance intensities of the significant DEPs normalized after unsupervised hierarchical clustering. **e**, MFuzz analysis showed that 68 out of 115 DEPs were upregulated during 10 organs development, while 47 DEPs were downregulated during organ development. **f**, Protein interaction network of the 115 DEPs. Network was visualized in Cytoscape and node colors are represented by Degree values. **g**, GO biological process and KEGG enrichment analysis results of the 115 DEPs. **h**, Organ-wise expression changes with age (within each column, from left to right) for the 9 spliceosome proteins in 10 organs co-expressing age-related DEPs. **i**, MFuzz analysis showed that all 9 DEPs were downregulated during 10 organs development. **j**, Dynamic expression changes of spliceosome protein RBM8A in brain (left) and U2AF2 (right). Statistical analysis used Student's t-test (**** p -value < 0.0001). In the box plot, the median is represented by the center line and the box boundaries represent the first and third quartiles.

3. Thanks to the reviewer for the question about whether we found any relevant/interesting effects on the protein sequence. Unfortunately, our current analysis cannot infer whether the sequence of the protein was affected. This is mainly because the DIA approach is bottom-up proteomics that is to say it is based on the characteristics (including sequence information) of the peptides to infer the proteins contained in the sample. If there is a change in the peptide, e.g., a loss or gain, this can lead directly to the detection of a different protein or a non-detectable protein. Protein sequencing and multiple digestion might be applied to the study of problems of peptide deletion or gain.

4. We truly thank the reviewer for the comments about supplementary data. We sent all supplementary data to the journal via a shared link after the first submission of the original article. We will ensure all the additional data will be re-uploaded to the submission system during re-submission process. At the same time, all mass spectrometry raw data have been uploaded to **ProteomeXchange** (a proteomics mass spectrometry data storage platform), and all analyzed data will be uploaded to both ProteomeXchange and **figshare** data sharing platforms for readers' access at the same time. ProteomeXchange reviewer account information is also available (Username: reviewer_pxd041400@ebi.ac.uk; Password: RQMHZ2YZ). If reviewers do not find the data, please feel free to let us know through the editors or any other means so that we can provide the data as soon as possible. Thanks.

- In a multi-batch experiment like this, removing batch effects is very important and influential on the findings. The authors acknowledge this step in the methods section (line 511) without providing any further details. Please elaborate. In addition, the methods seems to imply that each organ was measured as a “cohort”. Please elaborate on the measurement order, why this appears to be done in a per-organ basis and not (for statistical soundness) in a random order. How much does run-order influence the

clustering analysis? Please check on the HeLa measurements, whether the clustering observed there is largely driven by measurement order and if this does negatively affect the value of the resource.

1. We sincerely appreciate the detailed suggestion and inquiry about the batch effect from the reviewer. We would like to provide the following four aspects to confirm that the results of this study are not influenced by batch effects and that the conclusions are indeed reliable.

Firstly, for experimental manipulation, to ensure the stability of mass spectrometry analysis, we spiked iRT (indexed retention time), which is a mixture of 11 synthetic peptides that do not exist naturally, in the samples for analysis. The inclusion of iRT peptides enables the generation of high-quality spectral libraries and accurate normalization in DIA or SWATH detection, thereby enhancing the reliability of proteomic data and reduced the run-to-run variation (*Proteomics. 2012 Apr;12(8):1111-21.*).

Secondly, based on Čuklina et al.'s description of batch effects (*Mol Syst Biol. 2021 Aug;17(8): e10240.*), intensity boxplots are extremely powerful at the initial assessment stage. Simultaneously, with reference to the judgment methods proposed by Jiang et al. (*Cell. 2020 Oct 1;183(1):269-283.e19.*), we utilized log₂-transformed protein abundance data to determine the presence of batch effects in our cohort before conducting data analysis. No obvious discrete batch effects were observed in the intensity plot of protein abundances in each age group for each organ. Results added to the manuscript as supplementary Fig. 2 and we have also added descriptions from the manuscript:

Page 3, lines 104: Our protein abundance data, which were transformed using a logarithm base 2, demonstrate stability across different batches (Supplementary Fig. 2).

Page 13, lines 550: Before further analysis, protein abundances (in log₂ scale) in each organ type didn't shown discrete batch effects in this study (Supplementary Fig. 2).

Supplementary Fig. 2 | Protein abundances (in log₂ scale) in each age group for each organ. In the box plot, the median is represented by the center line and the box boundaries represent the first and third quartiles.

Finally, the newly supplemented mass spectrometry verification and WB validation experiments collectively confirmed our findings and illustrated the reliability of our analysis.

Page 7, lines 289: The instrument we used in this study was an EASY nLC 1200 coupled with QE HF-X system (Thermo Fisher Scientific). We further validated our brain-unique DEPs results on an additional timsTOF flex (Bruker) mass spectrometry platform. The results showed that both mass spectrometry detected 6,671 PGs in the brain, about 78% of the total amount (Supplementary Fig. 10a, Supplementary Data 15). Heatmaps and expression trends of the two mass spectrometry platforms regarding brain-unique age-related DEPs (ANOVA adjust p-value < 0.05) were shown in Supplementary Fig. 10b. The number of age-related down- and up-regulated DEPs detected by both platforms was 221 and 343, respectively (Supplementary Fig. 10c). PCA analysis also showed the consistency of the mass spectrometry data generated by the two different platforms (Supplementary Fig. 10d). Moreover, we generated quantile–quantile (Q–Q) plots⁵² to compare the P values reported of the data detected by the two mass spectrometers with the normalized rank P values, that represent the uniform distribution to illustrate the consistency of the data obtained by the two instruments (Supplementary Fig. 10e). The data points exhibit an evenly distributed pattern along the diagonal, indicating strong reproducibility of the detected data across both platforms.

52 Galwey, N. W. A Q–Q plot aids interpretation of the false discovery rate. *Biom J* 65, e2100309 (2023).

Page 5, lines 178: In our study, expression of RBM8A in the brain and U2AF2 in the kidney decreased significantly from infancy to adulthood (Fig. 2j). Western blot experiments confirmed the trend of decreasing expression of the splice variants with age, which is consistent with the mass spectrometry experiment (Supplementary Fig. 5). All original images can be viewed from ProteomeXchange.

2. The authors really appreciate your time and patience in helping us increase the readability and quality of this manuscript. In this study, we detected by organ type. We have additionally added a proteomics cohort containing 40 samples to validate whether the measurement order has any batch effect or influences the clustering analysis. These new results showed that the measurement order doesn't affect the final sample clustering (Supplementary Fig. 1e).

Supplementary Fig. 1e | Proteomic atlas of ten organs from infancy to adulthood mice. e, After order and disorder detection of a total of 40 samples from the brain, kidney, liver and spleen, the mass spectrometry results were analyzed by PCA.

The related contents are also given as below:

Page 3, lines 120: The multi-organ study also revealed clustering between different organs. To avoid run-order influence in the clustering analysis, we assessed whether the order of mass spectrometry detection would lead to batch effects. A total of 40 samples from brain, kidney, liver and spleen were performed a random mass spectrometry. We found that regardless of the order or disorder of LC-MS measurement, the samples tight clustered according to tissue types, indicating that in our study the detection order does

not affect the mass spectrometry data and cause batch effects (Supplementary Fig. 1e, Supplementary Data 5).

- The authors mention a DDA database and its analysis in the methods section (lines 480 and 495). However, this dataset is not mentioned anywhere. What does the dataset contain? What was its original purpose and why was it not used in the end?

Many thanks for the reviewer's inquiry. In our study, QC samples (e.g., sample mixtures and HeLa sample) are tested using the DDA method after the mass spectrometer instrument cleaning or calibration to assess instrument status. Sample mixtures were also assayed using the DDA method to assess pairwise column stability. We apologize for any confusion due to our lack of clarity, and we have revised the manuscript accordingly based on the reviewer's questions:

Page 12, lines 504: The DDA strategies were used for both quality control samples HeLa cell digested and tissue mixed samples to assess instrument status after the mass spectrometer cleaning or calibration.

- In e.g. lines 19 and 36, the sentence gives the wrong impression that samples for different time points were acquired from the same individual, which does not appear to be correct – see Methods. Please correct and tone down the wording in the abstract/introduction/discussion.

Thanks for the comments and we do apologize for the misunderstandings in the original manuscript. We have revised the phrase “self-multi-organs” to “multi-organ” throughout the manuscript to be more precise:

Page 1, lines 17: Herein, we present a comprehensive proteomic analysis of ten mouse organs (brain, heart, lung, liver, kidney, spleen, stomach, intestine, muscle and skin) at three crucial developmental stages (1-week, 4-week and 8-week after birth) by data-independent acquisition mass spectrometry.

Page 1, lines 25: This multi-organ proteome atlas provides a fundamental baseline for understanding the molecular mechanisms underlying organ development and maturation in early-life.

Page 2, lines 39: To date, the molecular profile of early-life multi-organ comparisons still remains scarce.

Page 2, lines 52: Mapping the multi-organ development proteome atlas can therefore detect the molecular regulators and effectors of developmental biology physiological activities.

Page 10, lines 400: This organism-wide characterization of early-life dynamic proteomics atlas fills in the knowledge gap for early-life multi-organ development and maturation.

Page 10, lines 402: Systematic and simultaneous detection of proteomic data from multiple organs under the same conditions could provide a more comprehensive picture of in vivo early life development.

Minor Issues

- Please consult a language office to improve overall readability.

Thanks for the suggestions. We have made meticulous modifications to the manuscript, aiming to present our findings in a more rigorous manner to the readers. Native speakers have also been employed to proofread the language.

- The authors did not provide the coverage of mouse proteome or the average protein coverage. In addition, the authors could add some more basic statistics on

Thanks for the reviewer's question. The DIA analysis software used in this study was DIA_NN (*Nat Methods*. 2020 Jan;17(1):41-44.), and after checking with the authors,

the DIA_NN did not calculate the protein coverage. I apologize for not being able to provide detailed information on peptide coverage.

The following information is provided for the reviewer's reference: The exported results showed a total of 200,245 identified peptides with different charges, of which 150,172 peptides were used to quantification of 11556 protein groups. The average amino acid length was 13. We found a total of 706 PGs containing a single peptide.

- The authors reported Pearson correlation coefficients (for example from line 115) without statistical significance test results, if they have done any. Considering the high dimensionality of the data, it may be beneficial to perform significance tests on the correlation measures.

We truly appreciate the suggestions, and we have done the statistical significance test of the Pearson correlation coefficient. We apologize for not providing the data to the reviewers in time and all additional data will be re-uploaded to the submission system during the re-submission process. The statistical significance test results of Pearson correlation coefficients are shown in the Microsoft OneDriver link below: <https://1drv.ms/x/s!AsvSofUEIDd3iVMsAhB4r3bfccgB?e=Ndj0KV>.

All the statistical test results will be uploaded to both ProteomeXchange and figshare data sharing platforms for readers' access at the same time.

- In the paragraph from line 110, the authors show the agreement of their data on single organs with already published single organ dataset in long, liver and stomach tissues (Supplementary Fig. 1c). However, as the authors mention in line 408, there are published data on the brain tissue. Why are the results not reported for this tissue?

We sincerely thank the reviewer for the detailed readings and inquiries. We hope to illustrate the reliability of our study by comparing it with other single-organ developmental studies. After reading the literature in details, we compare our data with

single-organ studies of the lung, liver, and stomach, which were conducted at a similar time point to allow for meaningful comparisons. As for the article on brain development, namely "A multiregional proteomic survey of the postnatal human brain", as cited in line 408 of the original manuscript, the sample spanned the period from early infancy (1 year after conception) to adulthood (42 years). However, due to the mismatch in time frame and animal species between that article and our study, a direct comparison of the data was not feasible. The purpose of including this citation was to highlight that both our study and other single-organ studies indicate the involvement of age-related differential proteins and genes in the splicing process and the spliceosome pathway.

- Since the data is generated in a DIA mode, the term “detected” is more suitable than “identified”. (e.g., line 21).

The authors greatly appreciate the suggestions. We do agree with the reviewer's suggestions regarding scientific descriptions and have accordingly made changes to the corresponding words throughout the revised manuscript.

- There are some inconsistencies and mistakes in the figure legends. For example, in Supplementary Figure 1, the legends of subfigures c and d are swapped.

We thank the reviewer for careful reading and pointing out this issue. We have modified the legends for Supplementary Figure 1 and also double checked the other figures.

Supplementary Fig. 1 | Proteomic atlas of ten organs from infancy to adulthood mice.

a, Summary of the sample age, organ type, number of mice, and total number of samples. **b**, Line plots depicting the alterations in the count of protein identifications across three different ages were generated for ten distinct organs. **c**, Dynamic Changes in housekeeping protein expression. In the box plot, the median is represented by the center line and the box boundaries represent the first and third quartiles. In the box plot, the median is represented by the center line and the box boundaries represent the first and third quartiles. $n=10$ biologically independent mice per organ per age. **d**, Venn diagrams show the comparison of the protein species identified compared to the results of other single-organ studies. **e**, After order and disorder detection of a total of 40 samples from the brain, kidney, liver and spleen, the mass spectrometry results were analyzed by PCA. **f**, Pearson correlation heatmaps of 10 organs.

- There needs to be more conclusions and discussions for some of the results that the authors present in the manuscript. For example in line 148 (Fig. 2e), the authors mention that some of the differentially expressed proteins can form a protein interaction network which they present in the Fig. 2e. However, there is no discussion/interpretation about this network.

We appreciate the reviewer for the valuable suggestions. In the revised manuscript, we have provided a further description of the results of the string analysis of the 115 DEPs and have added a discussion section. And as mentioned earlier, we have also discussed our results through additional mass spectrometry experiments and WB experiments.

Page 4, lines 162: 96 out of 115 DEPs could form a protein interactions network (Fig. 2f). KEGG enrichment results shown these inter-organ age-related proteins extensively engaged in metabolism of RNA. String analysis revealed that a remarkable 86 out of 96 proteins exhibited whole-body expression, implying that these 115 DEPs we propose have the potential to be applied to developmentally relevant differential proteins throughout the whole body from infancy to adulthood.

Page 10, lines 420: The biological functions of many proteins depend on specific physical interactions with other proteins, and define a complete map of protein-protein functional interactions in an organism has long been regarded as a pivotal objective in the post-genomic area^{62,63}. String analysis showed that 96 out of 115 DEPs could form protein interactions network, implying that these inter-organs age-related DEPs could be facilitated to defining the complete map of functional protein-protein interactions and certainly plausible results need to be verified by more detailed experiments.

62 Sahni, N. *et al.* Widespread macromolecular interaction perturbations in human genetic disorders. *Cell* 161, 647-660 (2015).

63 Wang, X. *et al.* Three-dimensional reconstruction of protein networks provides insight into human genetic disease. *Nat Biotechnol* 30, 159-164 (2012).

Page 10, lines 535: This study has also proposed organ-unique age-related DEPs, which can contribute to a better parsing the independent development of single-organ and the overall development of organisms. Further associations, these findings can

establish a bridge between individual organs and the organic whole, and are relevant for studies of organoids, organ-specific diseases, and systemic diseases. Interestingly, the inter-age correlation of the skin is different from other organs, and our proteomics data show that 4-week-old skin is closer to 1-week-old skin. This result is similar to that of Dekoninck et al. in murine tail and paw epidermis³⁴, both of which indicate that mouse skin undergoes rapid development until 4 weeks, after which it enters a period of maturation and stabilization. All of these phenomena pointed to commonalities in organ development as well as many organ-unique characteristics.

34 Dekoninck, S. *et al.* Defining the Design Principles of Skin Epidermis Postnatal Growth. *Cell* 181, 604–620. e622 (2020).

Fig. 2 | Age-related differential proteins across 10 organs from infancy to adulthood. f, Protein interaction network of the 115 DEPs. Network was visualized in Cytoscape and node colors are represented by Degree values.

• In line 152, the authors mention that since only spliceosome pathways were enriched in their data, they must be essential in development. This sentence shows overconfidence in the produced data and disregards the importance of other proteins and pathways that are not detected in this study.

We appreciate the reviewer’s detailed suggestions. Per the reviewer’s comment, we have revised the manuscript to describe the results using a more humble statement.

Page 4, lines 168: The fact that only one KEGG pathway, the spliceosome, had an adjusted p-value of fewer than 0.05 highlights the important role of spliceosome in organismal development and maturation (Fig. 2g).

- There are some occurrences of the term “significant” through the manuscript (for example, line 205) for which the authors did not provide the results of the statistical tests, if they have done any. If no statistical tests have been performed, I would suggest using other terms like remarkable and to avoid the term significant.

We appreciate the reviewer’s suggestions. Based on the reviewer’s comment, we have revised the manuscript by replacing "significant" with “remarkable” and “noteworthy” in statements that were not performed statistical analysis.

Page 2, lines 73: We showed that spliceosome proteins are not only extensively involved in the early-life development of multiple organs, but also have noteworthy modulatory functions for the expression of organ-unique properties and sexual dimorphism during early-life liver development.

Page 6, lines 229: For instance, the spleen had a **remarkable** decline in cell cycle proteins while the other three organs did not. The expression of proteins involved in oxidative phosphorylation in the spleen was not **noteworthy** increased with organ development.

Page 8, lines 314: We have found that some proteins showed temporal patterns and only expressed at a specific time.

Page 9, lines 372: This suggests that sex differences during the early-life development of the liver are the most remarkable among all 10 organs.

- In lines 150 and 154, the same paragraph about the spliceosome is repeated with different references. Please avoid repetitions.

We are extremely grateful to the reviewer for carefully pointing out our mistakes. We have deleted the repeated contents.

Page 5, lines 168: The fact that only one KEGG pathway, the spliceosome, had an adjusted p-value of fewer than 0.05 highlights the important role of spliceosome in organismal development and maturation (Fig. 2g). Spliceosome is mainly responsible for splice precursor messenger RNA (pre-mRNA), which is an essential step in the flow of information from DNA to protein in all eukaryotes⁷.

7 Mazin, P. V., Khaitovich, P., Cardoso-Moreira, M. & Kaessmann, H. Alternative splicing during mammalian organ development. *Nat Genet* 53, 925–934 (2021).

• The introduction mentions that DIA has no “bias to precursors in a pre-designed isolation ranges”. This is not true. Either remove or elaborate what was meant here.

We thank the reviewer for the comments and we apologize for not making the description clear. Per the reviewer’s comments, the original description of DIA is amended as below:

Page 2, lines 61: Instead of selecting specific precursors with higher intensities for fragmentation as in data-dependent acquisition (DDA), the mass spectrometer in DIA method is configured to cycle through a pre-defined set of precursor isolation windows, ensuring that all precursors within the desired mass range are consistently fragmented^{21,25,26}.

21 Demichev, V., Messner, C. B., Vernardis, S. I., Lilley, K. S. & Ralser, M. DIA-NN: neural networks and interference correction enable deep proteome coverage in high throughput. *Nat Methods* 17, 41–44 (2020).

25 Chapman, J. D., Goodlett, D. R. & Masselon, C. D. Multiplexed and data-independent tandem mass spectrometry for global proteome profiling. *Mass Spectrom Rev* 33, 452–470 (2014).

26 Ludwig, C. *et al.* Data-independent acquisition-based SWATH-MS for quantitative proteomics: a tutorial. *Mol Syst Biol* 14, e8126 (2018).

- It seems that the term “adulthood” is defined differently for mouse in comparison to the manuscript. Please elaborate why and 8-week mouse is considered adult and not a more classic 3-6 months definition was used.

Thanks for the comments. At the initiation of our study, we conducted a literature review on postnatal/early-life organ development in mice. We found that several studies used 6-8-week-old mice to represent adult mice. For example, Gour et al. investigated the postnatal development of the connectome in the mouse barrel cortex using mice aged 5, 7, 9, 14, 28, and 56 days, and the authors described this stage as equivalent to development from infancy to adulthood (*Science*. 2021 Jan 29;371(6528): eabb4534.). Li et al. used the stomachs of 8-week-old mice as adults for stomach developmental research (*Nat Commun*. 2018 Nov 21;9(1):4910.). Additionally, Moreira et al. et al. used 63-day postnatal mice as adults for their study (*Nature*. 2019 Jul;571(7766):505-509.). Based on the existing literature, we have chosen 8-week-old mice as the appropriate age group to define adult mice in our own study. If more detailed definitions of age are subsequently researched or discovered, we would be glad to adjust the description of "adulthood" accordingly.

- Figure 1 highlights 1300 samples, whereas 300 were measured/generated. Please correct or elaborate.

Thanks for the detailed suggestion. Based on the Reviewer’s comment, we have revised the Figure 1a to make it concise.

Fig. 1 | Proteome atlas of ten organs from infancy to adulthood mice. a, The workflow for DIA quantitative proteomics and bioinformatic analysis.

• In Figure 1c, it is very difficult to differentiate the organs/time points. Please check alternative visualizations.

We appreciate the suggestions. In the revised manuscript, we have included a protein ranking figure to accurately illustrate the highly dynamic nature of the proteome in our study. To highlight the dynamic ranking of each organ, we have reduced the size of the data points in the original plot and removed the age markers. Additionally, we have labeled the proteins with the highest and lowest values, and the raw data for protein ranking (Supplementary data2) is accessible for readers.

Fig. 1 | Proteome atlas of ten organs from infancy to adulthood mice. c, Dynamic ranges of the ten-organ proteomes.

• Figure 1e was generated with or without clustering on the samples? Do samples generally cluster as expected? Please also add color legend for the time points.

Thank you for the comment. Figure 1e displayed the Pearson correlation coefficients of all samples without clustering. We employed a Hierarchical Clustering approach, as illustrated in Supplementary Fig. 6a (shown below), to analyze all samples. The clustering results revealed that muscle tissue exhibited greater distinctiveness compared to other tissues, while the remaining samples clustered according to organ type, as anticipated. Moreover, a few organs displayed a tendency to cluster based on age, such as the 1-week stomach, kidney, and lung.

Supplementary Fig. 6a | Age-related differential proteins across 9 organs from infancy to adulthood. a, Hierarchical clustering of 300 samples.

Following the reviewer's suggestion, we have revised Figure 1e and a color legend for the time points.

Fig. 1e | Proteome atlas of ten organs from infancy to adulthood mice. e, The Pearson correlation coefficient of all samples.

• Supplementary Figure 6 is not annotated at all and thus the clustering shown does not “show” anything. Please add some form of additional annotation.

Many thanks for the specific comment. We have included organ labels in the revised manuscript figure. This addition aims to enhance the visualization of clustering patterns among organs of the same age. The resubmitted version of Supplementary Figure 6 becomes Supplementary Figure 9.

Supplementary Fig. 9 | Pearson correlation coefficient of ten organs proteomic data from infancy to adulthood. a, 1 week. b, 4-week. c, 8-week.

- Please add a colored label in Supplementary Figure 1 for gender. Also, why is the profile of e.g. heart, intestine and liver so different from the rest?

We truly appreciate the reviewer's comments. Per the reviewer's comment, we have made modifications to Supplementary Figure 1 by adding gender information labels. Additionally, we have adjusted the color scheme of the heatmap to minimize any visual discrepancies caused by strong chromatic aberration. The heart is probably biased due to the bias caused by the color assigned to the heat map. The intestine is mainly a cause of individual differences. And the liver exhibited sex dimorphism. Modifying the colors allows for a clearer highlighting of tissue correlations across different ages.

Supplementary Fig. 1 | Proteomic atlas of ten organs from infancy to adulthood mice.

a, Summary of the sample age, organ type, number of mice, and total number of samples. **b**, Line plots depicting the alterations in the count of protein identifications across three different ages were generated for ten distinct organs. **c**, Dynamic Changes in housekeeping protein expression. In the box plot, the median is represented by the center line and the box boundaries represent the first and third quartiles. **In the box plot, the median is represented by the center line and the box boundaries represent the first and third quartiles. n=10 biologically independent mice per organ per age.** **d**, Venn diagrams show the comparison of the protein species identified compared to the results of other single-organ studies. **After order and disorder detection of a total of 40 samples from the brain, kidney, liver and spleen, the mass spectrometry results were analyzed by PCA.** **f**, Pearson correlation heatmaps of 10 organs.

• Supplementary Figure 2 may point to the presence of sample mixups? E.g. Intestine 8-week female and muscle 4-week female. Please double check that all 300 samples are correctly labeled.

We sincerely appreciate the reviewer's comments and inquiries about Supplementary Figure 2 (now realigned as Supplementary Figure 4). To avoid any issues arising from confusion in sample preparation or data analysis, we thoroughly examined the experimental records and identified no potentially problematic instances that could indicate sample mix-ups. We speculate that the observed variations may be attributed to individual differences in the samples. Additionally, considering the larger muscle and

intestinal tissues, it is possible that the discrepancies are influenced by the spatial location of the samples themselves. After excluding these four samples (intestine_8w_F-1, muscle_4w_F-1&2&4), we observed no impact on the clustering of organ ages, thus indicating that the final results remain plausible.

Overall, we would like to express our gratitude again to all the reviewers for their time in reading and facilitating our manuscript. We sincerely appreciate the insightful comments and suggestions. We hope our additional experimental data and elaborations may adequately address the reviewers' questions and inquiries.

REVIEWER COMMENTS

Reviewer #2 (Remarks to the Author):

I have no additional questions or concerns with the revised version of the manuscript and would support publication at this point.

Reviewer #3 (Remarks to the Author):

The authors have extensively worked on the manuscript, added new data, new figures, new analysis. I commend the authors for their diligence. Some minor points remain, connected to what I have requested earlier:

- The authors mention that the analysis of the effects of spliceosome on the protein sequences is not possible since they have used a bottom-up DIA approach. However, I do not agree with this statement. It is generally feasible to demonstrate differences in detected peptides at various time points, shedding light on potential effects of the spliceosome. For instance, analyzing peptides leading to the inference of different gene products (e.g. isoforms) can be used to show differential splicing. If the authors simply do not find any peptides connected to e.g. different isoforms, then this could be added to the text.
- The authors have added some analysis to support the lack of batch effects. I recommend extending this by e.g. simply adding a single PCA with all samples (no filtering on samples or proteins) to show that samples cluster as expected. For the QC samples, a simple plot showing e.g. the number of proteins or peptides could be added to show that no effect along measurement time on instrument performance was observed.
- The calculation of protein sequence coverage and overall protein coverage of the mouse proteome can be easily performed, and given the extensive analysis the authors have done on the other data, I would expect them to be able to calculate this even though DIA-NN does not report such metric itself. I view this as a very minor point.

RESPONSES TO REVIEWER COMMENTS (NCOMMS-23-17406B)

The authors would like to express our gratitude to *Reviewer#2* for approving the publication of our work in *Nature Communications*. Additionally, we sincerely thank *Reviewer #3* for the insightful and valuable comments on improving the quality of our paper. Accordingly, we have provided point-to-point responses and made requested modifications to the manuscript.

Reviewers' comments are in blue, with authors' responses immediately following each comment and in black. Original text modified with a yellow background.

Reviewer #3's comments:

The authors have extensively worked on the manuscript, added new data, new figures, new analysis. I commend the authors for their diligence. Some minor points remain, connected to what I have requested earlier:

- The authors mention that the analysis of the effects of spliceosome on the protein sequences is not possible since they have used a bottom-up DIA approach. However, I do not agree with this statement. It is generally feasible to demonstrate differences in detected peptides at various time points, shedding light on potential effects of the spliceosome. For instance, analyzing peptides leading to the inference of different gene products (e.g. isoforms) can be used to show differential splicing. If the authors simply do not find any peptides connected to e.g. different isoforms, then this could be added to the text.

We sincerely appreciate the valuable suggestions and apologize for the misunderstanding from our previous responses. We analyzed our data and compared the results with the Uniprot database, and no peptides were found to be connected to different isoforms. We speculate that this may be due to the current library-based proteomics approach, which lacks the precision to accurately distinguish the differences of several dozen or even a few amino acids loss or gain between isoforms. Additionally, it is also possible that the protein databases currently used do not explicitly present the amino acid sequences of different isoform proteins separately as independent search items, resulting in the inability to distinguish different isoform information during library search. We highly agree with the idea proposed by the reviewer to infer the occurrence of splicing events based on isoform identification at different time points. In our ongoing second study on multi-organ aging in mice, apart from proteomics, we have also conducted transcriptomics analysis. According to the suggestions of the reviewers, we will further investigate the changes in different protein isoforms and splicing during the process of multi-organ aging, with the support of a dual-omics approach.

Per the reviewer's comment, we have also revised the manuscript accordingly:

Page 11, lines 436: Thousands of protein isoforms are derived from splicing changes, and these isoforms are linked to crucial physiological processes such as cell

differentiation, organ development, and tissue homeostasis⁶⁶. By comparing the different protein isoforms identified at different stages, differences in splicing events might be inferred. Further research might be conducted by optimizing the detection accuracy of proteomics or utilizing transcriptomics and protein sequencing to address the relevant issues^{7,67}.

7 Mazin, P. V., Khaitovich, P., Cardoso-Moreira, M. & Kaessmann, H. Alternative splicing during mammalian organ development. *Nat Genet* 53, 925–934 (2021).

66 Baralle, F. E. & Giudice, J. Alternative splicing as a regulator of development and tissue identity. *Nat Rev Mol Cell Biol* 18, 437–451 (2017).

67 Singh, A. *et al.* Broad misappropriation of developmental splicing profile by cancer in multiple organs. *Nat Commun* 13, 7664 (2022).

- The authors have added some analysis to support the lack of batch effects. I recommend extending this by e.g. simply adding a single PCA with all samples (no filtering on samples or proteins) to show that samples cluster as expected. For the QC samples, a simple plot showing e.g. the number of proteins or peptides could be added to show that no effect along measurement time on instrument performance was observed.

We truly appreciate the reviewer for the insightful comments. Based on the reviewers' suggestions, we have made further improvements to the manuscript. Firstly, we performed PCA analyses on no filtering data. Secondly, we included the number of proteins identified by QC samples. Lastly, we have reorganized and discussed the results section regarding batch effects.

Page 3, lines 102: The correlation between the biologically replicated samples of organs of the same sex averaged 0.974, which indicates the stability of the data (Fig. 1d Supplementary Data 3). To check for batch effects or run-to-run variations, we performed the following analyses. An initial PCA analysis was conducted on all samples without any sample or protein filtering. As expected, the samples exhibited clustering patterns based on organ (Supplementary Fig. 1c). According to previous studies on batch effects by Jelena *et al.*³², we also examined the data after performing a log₂ transformation and found no significant differences among the batches (Supplementary Fig. 1d). The number of proteins identified by QC was stable, showing that no effect along measurement time on instrument performance was observed (Supplementary Fig. 1e). Moreover, in order to investigate the potential impact of the sequencing order in mass spectrometry studies, a total of 40 samples from the brain, kidney, liver, and spleen were analyzed using random mass spectrometry. We found that regardless of the order or disorder of LC-MS measurement, the samples tightly clustered according to tissue types, indicating that in our study the detection order does not affect the mass spectrometry data or cause batch effects (Supplementary Fig. 1f, Supplementary Data 5). The coefficient of variation (CV) is used to assess the relative variability of the dataset, with higher CV values representing more dispersion³³. In our study, protein quantification in most biological replicate samples showed median

coefficient of variance (CV) below 0.3 (Supplementary Fig. 3, Supplementary Data 4). Due to individual variation, the intestinal and muscle samples exhibited slightly higher coefficients of variation (CVs).

To further verify the accuracy of our proteomic data, we first verified randomly selected proteins from the Housekeeping gene pool³⁴ and confirmed that they were stably expressed across time and organs (Supplementary Fig. 1g). We also aimed to compare our findings with other existing studies on the proteomes of mouse single-organ¹¹⁻¹³.

- 11 Gong, T. et al. A time-resolved multi-omic atlas of the developing mouse liver. *Genome Res* 30, 263-275 (2020).
- 12 Li, X. et al. A time-resolved multi-omic atlas of the developing mouse stomach. *Nat Commun* 9, 4910 (2018).
- 13 Moghieb, A. et al. Time-resolved proteome profiling of normal lung development. *Am J Physiol Lung Cell Mol Physiol* 315, L11-L24 (2018).
- 32 Čuklina, J. et al. Diagnostics and correction of batch effects in large-scale proteomic studies: a tutorial. *Mol Syst Biol* 17, e10240 (2021).
- 33 Botta-Dukát, Z. Quartile coefficient of variation is more robust than CV for traits calculated as a ratio. *Sci Rep* 13, 4671 (2023).
- 34 Hounkpe, B. W., Chenou, F., de Lima, F. & De Paula, E. V. HRT Atlas v1.0 database: redefining human and mouse housekeeping genes and candidate reference transcripts by mining massive RNA-seq datasets. *Nucleic Acids Res* 49, D947-d955 (2021).

Supplementary Fig. 1 | Proteomic atlas of ten organs from infancy to adulthood mice. **a**, Summary of the sample age, organ type, number of mice, and total number of samples. **b**, Line plots depicting the alterations in the count of protein identifications across three different ages were generated for ten distinct organs. **c**, PCA analyses were performed on proteomic data from all ten organs without any filtering on samples or proteins. The samples clustered based on organ type but not by batch. **d**, Protein abundances (in log₂ scale) in each age group for each organ. In the box plot, the median is represented by the center line and the box boundaries represent the first and third quartiles. **e**, The number of proteins identified by QC was stably above 5000, showing that no effect along measurement time on instrument performance was observed. **f**, PCA visualization was performed on a total of 40 samples from the brain, kidney, liver, and spleen after conducting ordered and unordered detection. **g**, Dynamic Changes in housekeeping protein expression. In the box plot, the median is represented by the center line and the box boundaries represent the first and third quartiles. In the box plot, the median is represented by the center line and the box boundaries represent the first and third quartiles. n=10 biologically independent mice per organ per age. **h**, Venn diagrams show the comparison of the protein species identified compared to the results of other single-organ studies.

- The calculation of protein sequence coverage and overall protein coverage of the mouse proteome can be easily performed, and given the extensive analysis the authors have done on the other data, I would expect them to be able to calculate this even though DIA-NN does not report such metric itself. I view this as a very minor point.

Thanks for the reviewer's suggestions. According to the reviewer's request, we have performed the following additional calculations. Firstly, we performed peptides alignment by BLASTP. When multiple peptides were identified within one protein, we removed 100% identity duplicates to avoid recalculations. Then, for each protein, we calculated the length and number of matched peptides, revealing an overall protein coverage of 22.865%. Finally, the results of each protein sequence coverage have been uploaded to the supplementary data1 and the frequency distribution of peptide coverage is shown below:

We would like to thank Reviewer #3 again for your time and help to facilitate our manuscript.